# Rapid and accurate prediction of protein homo-oligomer symmetry using Seq2Symm

Meghana Kshirsagar [1] ✉, Artur Meller [2,3,13], Ian R. Humphreys [4,5,13], Samuel Sledzieski [1,6], Yixi Xu[1], Rahul Dodhia[1], Eric Horvitz[7,8], Bonnie Berger[6,9], Gregory R. Bowman [10], Juan Lavista Ferres [1], David Baker [4,5,11] & Minkyung Baek [12] ✉

The majority of proteins must form higher-order assemblies to perform their biological functions, yet few machine learning models can accurately and rapidly predict the symmetry of assemblies involving multiple copies of the same protein chain. Here, we address this gap by finetuning several classes of protein foundation models, to predict homo-oligomer symmetry. Our best model named Seq2Symm, which utilizes ESM2, outperforms existing template-based and deep learning methods achieving an average AUC-PR of 0.47, 0.44 and 0.49 across homo-oligomer symmetries on three held-out test sets compared to 0.24, 0.24 and 0.25 with template-based search. Seq2Symm uses a single sequence as input and can predict at the rate of ~80,000 proteins/ hour. We apply this method to 5 proteomes and ~3.5 million unlabeled protein sequences, showing its promise to be used in conjunction with downstream computationally intensive all-atom structure generation methods such as RoseTTAFold2 and AlphaFold2-multimer. Code, datasets, model are available at: https://github.com/microsoft/seq2symm.

Across nature, proteins often form assemblies involving multiple subunits to perform their biological functions. When multiple identical protein subunits are held together by non-covalent interactions, the resulting protein complex is called a homo-oligomer. Homo-oligomers can range in size from dimers, which have two identical subunits, to large oligomeric complexes with hundreds of subunits. Homo-oligomerization can be essential for the proteins' stability, folding, and function. For instance, some enzymes require the formation of a homo-oligomer to recognize their substrates[1].

The global arrangement of the identical subunits (>=95% sequence identity over 90% of the length of the subunits) in a homo-oligomer defines their symmetry. This can be either *point group symmetry*, involving the placements of subunits along one or more axes of rotation, or a *helical symmetry*, which involves both rotation and translation of the subunits along the axis of rotation[2]. The most common type of point group symmetry is cyclic ($C_n$ symmetry) where the complex consists of $n$ subunits rotated around a central axis. For example, this type of symmetry is often found in membrane proteins[3] which require a central pore, such as the β-Barrel pore-forming toxins (β-PFT), a large family of bacterial toxins[4]. Another common point group symmetry is dihedral symmetry ($D_n$ symmetry), in which homo-oligomers contain both a rotational axis of symmetry and

[1]AI for Good Research Lab, Microsoft Corporation, Redmond, WA, USA. [2]Department of Biochemistry and Molecular Biophysics, Washington University in St. Louis, St. Louis, MO, USA. [3]Medical Scientist Training Program, Washington University in St. Louis, St. Louis, MO, USA. [4]Department of Biochemistry, University of Washington, Seattle, WA, USA. [5]Institute for Protein Design, University of Washington, Seattle, WA, USA. [6]Computer Science and Artificial Intelligence Laboratory, Massachusetts Institute of Technology, Cambridge, MA, USA. [7]Microsoft Corp, Redmond, WA, USA. [8]Stanford Institute for Human-Centered Artificial Intelligence, Stanford, California, USA. [9]Department of Mathematics, Massachusetts Institute of Technology, Cambridge, MA, USA. [10]Department of Biochemistry and Biophysics, University of Pennsylvania, Philadelphia, PA, USA. [11]Howard Hughes Medical Institute, University of Washington, Seattle, WA, USA. [12]Department of Biological Sciences, Seoul National University, Seoul, South Korea. [13]These authors contributed equally: Artur Meller, Ian R. Humphreys. ✉e-mail: meghana.kshirsagar@microsoft.com; minkbaek@snu.ac.kr

perpendicular axes of two-fold symmetry. Dihedral symmetry is common among cytoplasmic enzymes because it facilitates a variety of protein-protein interfaces, enabling allosteric control[2]. In addition, homo-oligomers may adopt a cubic symmetry that combines 3-fold rotational axes with other non-perpendicular rotational axes such as icosahedral symmetry seen in viral capsules.

Despite the importance of homo-oligomerization for protein function, predicting the quaternary state and symmetry group of a protein given a single chain remains challenging. Currently, annotations of oligomeric states in the Protein Data Bank (PDB) are based on predictions from the PISA algorithm[5,6], supplemented by the assignments made by the researchers who deposit the structure. Although PISA is recognized for its high accuracy[6], this method relies upon an experimentally determined structure to extract assembly information and inform the most likely oligomeric state.

Methods that predict oligomeric state without experimental data often rely on homology template searches (such as HHSearch[7]) against known assemblies or employ docking-based symmetric transformations of monomers to model complexes[8]. One such method, GalaxyHomomer[9], combines template-based and docking-based approaches, and incorporates loop refinement to improve structure prediction. Recently, as a result of methods for highly accurate protein structure prediction[10,11], AlphaFold has been shown to predict homo-dimers at a proteome-scale, and in select cases higher order oligomeric assemblies[12]. However, using AlphaFold[11] or RoseTTAFold[10] for ab initio oligomeric state prediction poses significant computational challenges, as it requires running inference for each potential number of chains to score various copy number models, and is generally limited to proteins with high-quality MSAs.

More computationally efficient methods to fold large protein oligomers, such as Uni-Fold Symmetry[13] still require the pre-specified symmetry group as input to make predictions. MoLPC2[14] presents an approach that can algorithmically determine symmetry by building a multi-chain structure iteratively using a Monte Carlo Tree Search (MCTS) approach, starting from a single chain and scoring all the possible intermittent combinations of chains using AlphaFold-Multimer. This is, however, an inefficient approach as it requires running the five different AlphaFold-Multimer models for every assembly attempted in the MCTS. Protein embeddings from ESM2[15] have been used to predict the most likely quaternary state of a protein chain (QUEEN[16]); however, in this approach, the model only predicts the multiplicity of the oligomer thereby giving no clue as to global symmetry of the protein.

We set out to fine-tune protein foundation models (pFMs) to predict homo-oligomer symmetry. We define as *pre-trained* any approach that involves a protein model being used as a feature extractor feeding a classifier model (for instance, a logistic regression or neural network classifier) which is then trained on homo-oligomer symmetry prediction. We use *fine-tuning* to refer to any approach that involves modifying any parameters from the protein model by training them explicitly for oligomer symmetry prediction. Our approach, outlined in Fig. 1, can be applied to diverse protein families and its rapid runtime enables proteome-scale annotations.

Here, we show results from fine-tuning ESM2, ESM-MSA and RoseTTAFold2, as well as a baseline using template-matching for homo-oligomer symmetry prediction. We evaluate various approaches quantitatively on three different test sets, thereby providing broad support for the performance obtained by our approach. We experiment with both sequence and MSA as input representations to the methods we explore. We further provide rigorous evaluation of our best-performing model, which we call Seq2Symm, under different application scenarios involving varying levels of homology thereby giving insights into the capabilities and limitations of our approach. We illustrate the speed of Seq2Symm with large-scale evaluation on multiple proteomes and find that the results align qualitatively, with

prior work. Finally, we show how Seq2Symm can be used with AlphaFold2-multimer to generate homo-oligomer structures and show the significant computational gains as compared to a brute-force search-based approach using AF2-multimer. In addition to these results, our analysis of the homo-oligomer symmetry data from PDB, provides valuable information for future work.

## Results

We evaluate the various methods on our PDB-derived benchmarking dataset consisting of 129,013 structures which are split into training, validation and test splits in a sequence-aware manner (30% sequence identity over 80% coverage is used to define sequence-similar proteins), to restrict the similarity between the training and test splits. We call this data split a 'conventional split', since the same criterion has been used in prior work to define data splits in deep learning models such as AlphaFold[11], DeepMSA[17], DeepTMHMM[18]. A pairwise sequence identity of 30% is at the threshold of the "twilight zone" of homology, as prior work[19] found that among protein pairs with less than 25% sequence identity, fewer than 10% were homologous. We use three additional datasets for evaluation: a UniFold test set[13], a dataset curated from the latest PDB homo-oligomers ("PDB 2024"), and a de novo set of proteins (see Methods and the Supplementary material for details).

### An ESM2 fine-tuned model outperforms other approaches

We evaluated pre-trained and fine-tuned variants of ESM-MSA[20] (Evolutionary Scale Modeling using Multiple Sequence Alignments), ESM2[15] (Evolutionary Scale Modeling 2) and RoseTTAFold2[21] (RF2) against a template-based method, HHSearch[7], using metrics suited for class-imbalanced datasets and multi-label classification: Area Under Precision-Recall Curve (AUC-PR), confusion matrices, F1-scores, and Precision-Recall curve plots (detailed results in Supplementary Tables 1–3 and Supplementary figs. 1–4). We experiment with fine-tuning a varying number of layers in the pFM and try different feed-forward neural network architectures for the classifier head block shown in Fig. 1 (see Methods).

We find that an ESM2 fine-tuned model (which we call Seq2Symm) with a modified version of the language modeling head used in RoBERTa, trained with margin loss (see Methods) performs the best on all datasets (Fig. 2a,b), with further improvements seen with an expanded training dataset (i.e. with distillation). The next best macro averaged AUC-PR is obtained by the ESM2 pre-trained model with 0.47, 0.40, 0.38 and 0.24 on the validation, test, UniFold and PDB-2024 datasets, respectively, while ESM-MSA fine-tuned and RF2 fine-tuned models perform worse (test AUC-PR of 0.36 and 0.34 respectively). In comparison, the template-based HHSearch method achieved a far lower performance than most of the pFMs with a AUC-PR of 0.34, 0.24, 0.24 and 0.25 on the validation, test, UniFold and PDB-2024 datasets, respectively.

Next, we interrogated the strengths and weaknesses of Seq2-Symm, by looking at class-wise AUC-PR (Fig. 2d and class distribution of the test set shown in Fig. 2g). We find that it accurately identifies proteins across most cyclic symmetries (except C7–C9), across D symmetries such as D2, D3, D5, as well as helical and icosahedral symmetries. Looking at the confusion matrices, Seq2Symm has a lower tendency to overpredict the majority C1 and C2 class (Fig. 2c; bottom), unlike many other models (Supplementary fig. 4) and compared to the template-based method HHSearch (Fig. 2c; top), it is more likely to correctly predict dihedral, higher order cyclical, helical, and icosahedral symmetry groups. The only symmetries it performs worse on are 'O' (octahedral) and 'T' (tetrahedral). Some confusion categories are more easily rationalized: C4 confused as D2, C10–C17 are confused as helical, C7–C9 are confused as D6–D12 (likely due to the co-occurrence of some higher-order C and D symmetry groups, Supplementary Table 4), while some others are unclear: O confused as C6.

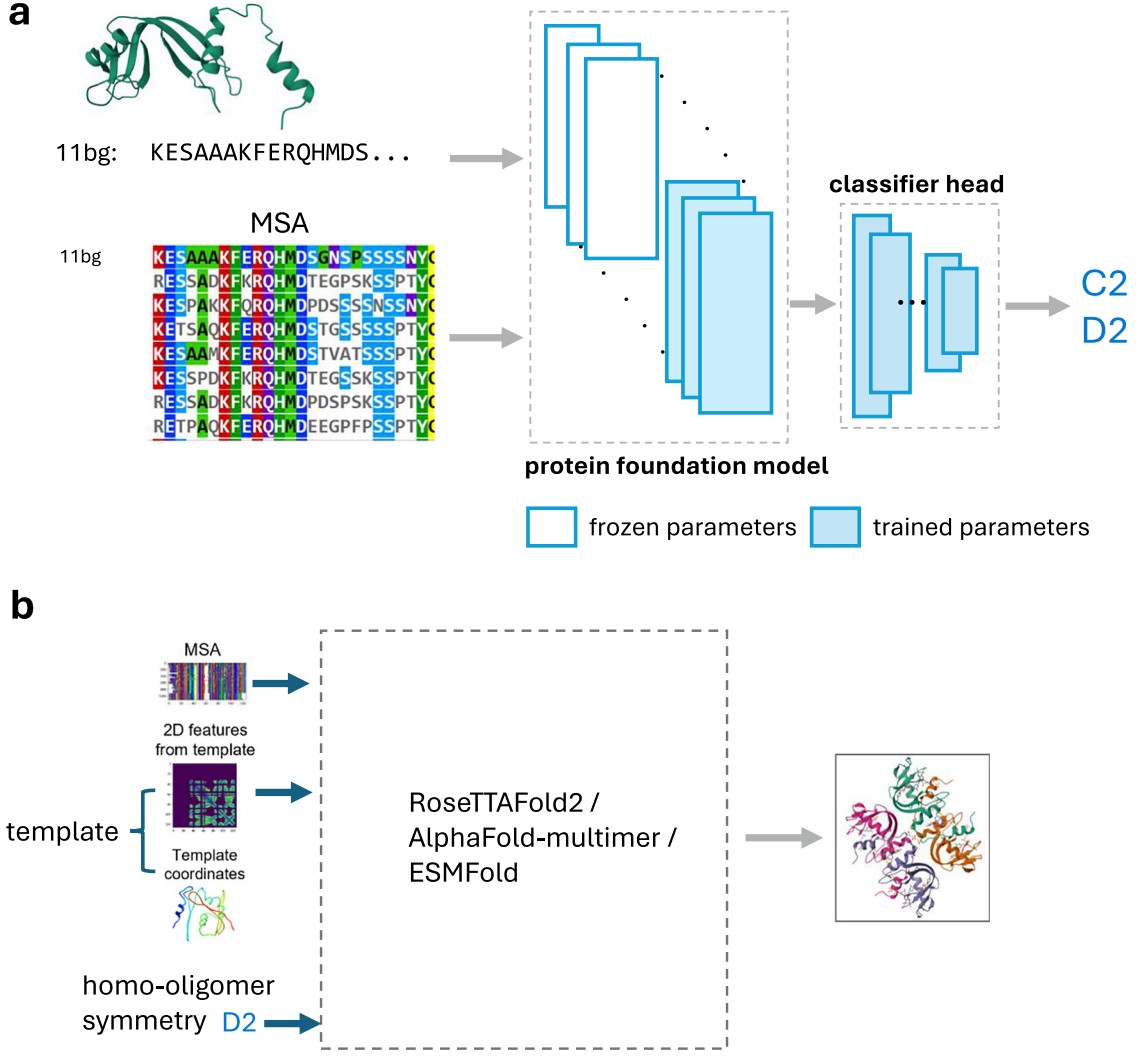

Homo-oligomer symmetry predictions are used as input to 3D structure predictors

**Fig. 1 | Protein foundation models can be fine-tuned to predict a protein's homo-oligomer symmetry. a** Schematic showing our modeling setup for multi-label prediction of homo-oligomer symmetry, illustrated for the bovine seminal ribonuclease protein (PDB id: 11bg). The input can be either the protein amino-acid sequence and/or the multiple sequence alignment (MSA). The 'protein foundation model' (pFM) can be ESM-MSA, ESM2, or RoseTTAFold2 (RF2). We experiment with various architectures for the 'classifier head' (see Methods). We vary the number of layers we fine-tune in the pFM, from a fully frozen model with a single trainable prediction head (i.e., "pre-trained only") to a model with all weights freely tunable (i.e., "fine-tuned"). **b** The homo-oligomer symmetry prediction can then be supplied to a structure prediction algorithm (e.g., AlphaFold, RoseTTAFold2, or ESMFold) to guide the generation of an atomic-resolution homo-oligomer structure.

## Understanding the strength of sequence-based modeling

We observed that ESM2-based models outperform RF2 and ESM-MSA (Fig. 2a) and analyze this difference further in Fig. 2e, where we average the performance of the various sequence-based models and compare it with the average of the various MSA-based models. We find that the sequence-based models (blue bars) outperform the MSA-based models (gray bars) on all oligomer symmetries except 'O' and 'I'. To assess the effectiveness of sequence-based models, we construct another training regime that features no homology between examples in the train and validation/test splits (e-value < 0.1; see Supplementary Table 5), that we call the 'no-homology' split, which is relevant in applications where de novo proteins are encountered, such as in protein design. Interestingly, we find that while ESM2-based models still marginally outperform other methods on this split, the difference between methods and the overall performance significantly decreases (Supplementary figs. 5–6). This, along with model performance on de novo designed proteins (Supplementary fig. 7), indicates that pLM approaches struggle on proteins without homology to the training dataset.

This prompted us to examine the relationship between oligomeric symmetry in our dataset and the broad notion of protein similarity, in particular, how the latter impacts the former. Investigating through the lens of protein family annotations, we find that proteins from the same Pfam family can have different homo-oligomer symmetry. For instance, the protein family PF00072 has proteins spanning the following symmetries: C6, C2, D2, T, H, D3. This is true for 3621 PFam protein families, representing 50% of all PFam families in our full dataset, where two proteins from the same family have a different homo-oligomer symmetry (Supplementary fig. 8). In the remaining 50% of PFam families, all proteins have the same symmetry. These constitute ~14% of the structures in our dataset and are largely (57%) monomers (C1).

Stratifying Seq2Symm's performance at the protein family level, based on protein families that were 'seen' or 'unseen' during training, we observe that out of 589 unseen protein families, Seq2Symm has 100% recall and 100% precision, respectively, on 276 and 259 unseen families (Supplementary fig. 9c,d). This illustrates some ability to

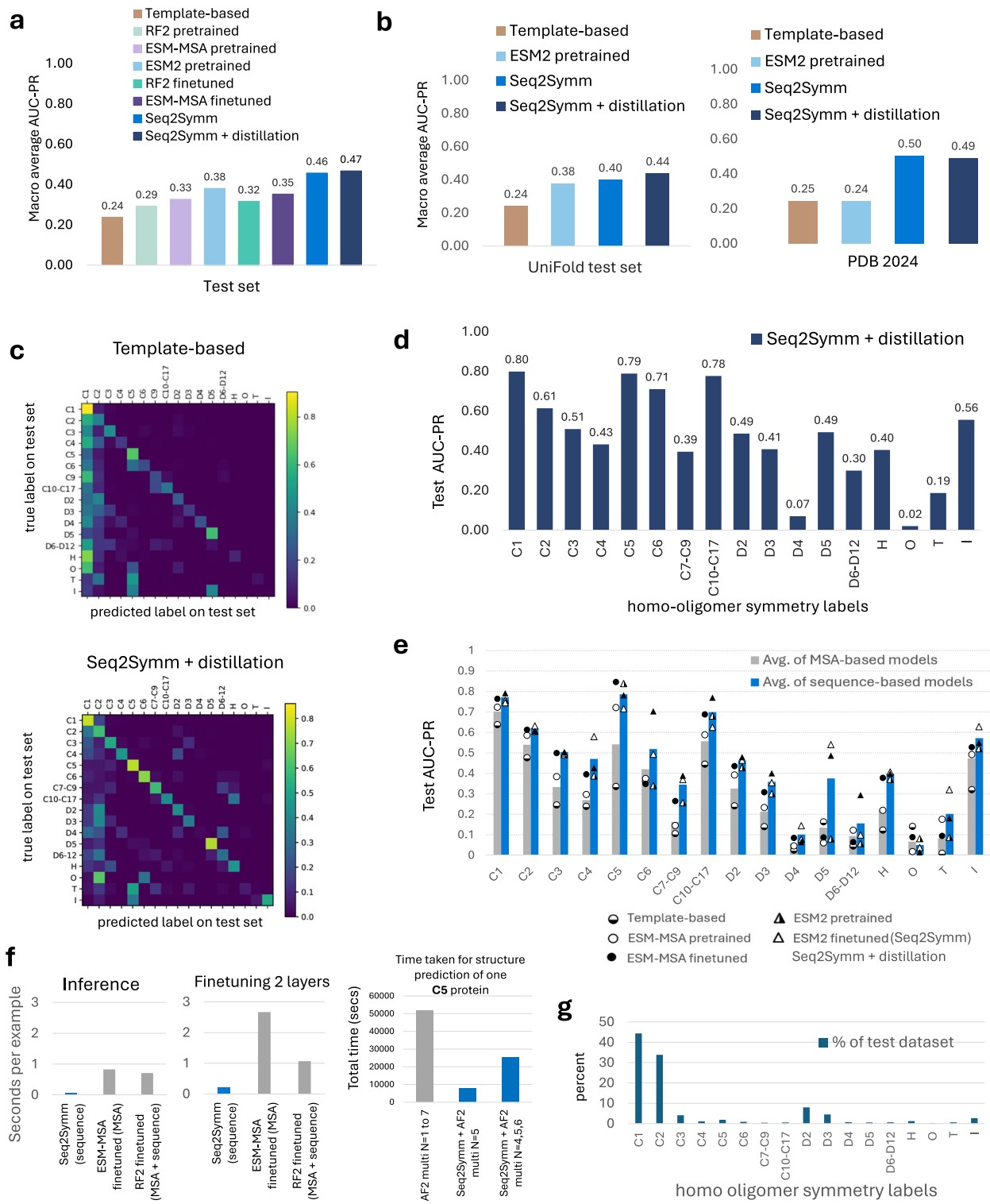

generalize to less homologous proteins (see Supplementary Table 6a for detailed results, Supplementary Table 6b for CATH-stratified analysis).

Moving to the coarser level of MSAs (Supplementary fig. 10a,b), where we analyze the diversity of predicted oligomer symmetries over all proteins within an MSA, and a qualitative look at orthologous proteins picked from some of these MSAs (Supplementary fig. 11), we notice that similar proteins can adopt different oligomeric symmetries in different organisms. These diverse oligomeric symmetries may contribute significant noise in MSA-based methods that rely on co-

evolution based representations which can group more distant proteins.

Finally, to get a fair assessment of template-based methods, we design a setting to simulate the "typical" manner of applying HHSearch-like methods that are not "trained". Using a sequence identity threshold of 95% at a coverage of 90%, we create a data-split where the test set does not have identical or near-identical structures to the training set. We evaluate both HHSearch and Seq2Symm (a model trained on this "95% seq-id" training split) on this test set. We find that HHSearch has a macro-averaged test AUC-PR of 0.542 with

**Fig. 2 | Protein foundation models predict homo-oligomer symmetry more accurately than current template-based methods. a** Performance, measured using area under the precision-recall curve (AUC-PR), for the various methods on the held-out test split of our dataset. The AUC-PR shown is the macro-average over class-wise AUC-PR, with class-weighted AUC-PR results as well as validation set results in Supplementary fig. 1. **b** Performance of representative models on two other completely unseen datasets, the "UniFold test set" from prior work (see Methods for dataset details, Supplementary Table 7) and "PDB 2024" involving homo-oligomers released by PDB in 2024 (see Supplementary Table 13 for details). The AUC-PR is a macro-average over class-wise AUC-PR for the classes in this dataset. **(c)** Confusion matrix of one of the baselines: HHSearch and Seq2Symm (a fine tuned ESM2-based model), showing the symmetries where there is confusion. This matrix is shown for only proteins with a single label (i.e. multi-label examples are excluded). **d** Test AUC-PR for each homo-oligomer symmetry shown for the best model, an ESM2-based fine tuned model. **e** Class-wise AUC-PR on the test set, averaged over sequence-based models (orange bars) and MSA-based models (blue bars) in the bar chart, with individual model performances in each category shown by the points. We find that the models using a sequence-only representation (triangle points, $n = 3$) achieve a higher AUC-PR for nearly every symmetry class, as compared to the MSA representation-based models (circular points, $n = 3$). The biggest gains are seen on higher-order symmetries such as C4, C5, C7–C9, D5. **f** Inference and training time taken by each protein foundation model, shown per input example. The rightmost plot compares the time taken for full structure prediction using a brute-force search involving AF2 multimer vs. an approach that uses Seq2Symm to obtain the homo-oligomer symmetry first followed by structure generation using AF2. Total time is shown in seconds averaged over 10 proteins with C5 symmetry. **g** Distribution of homo-oligomer symmetries in our test set.

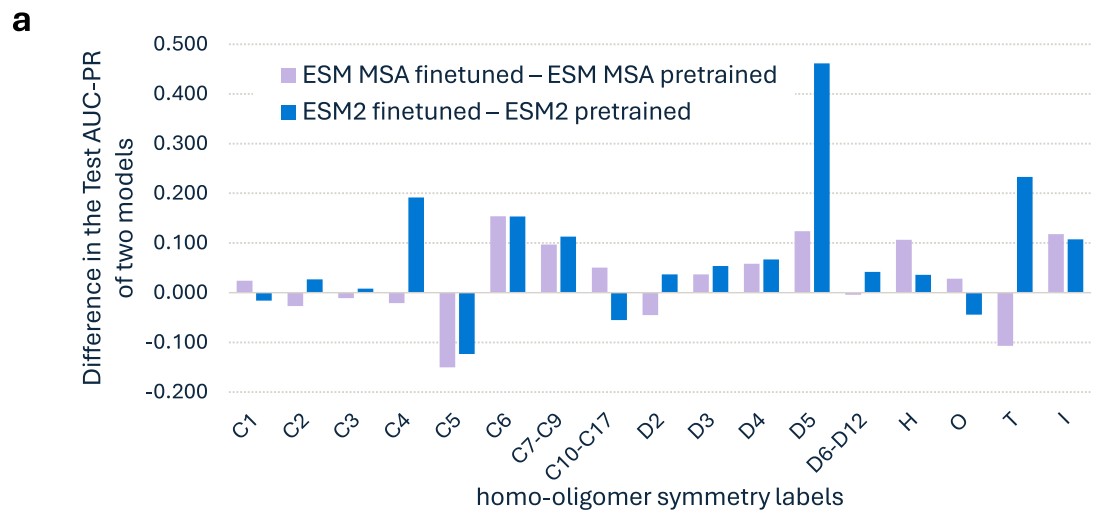

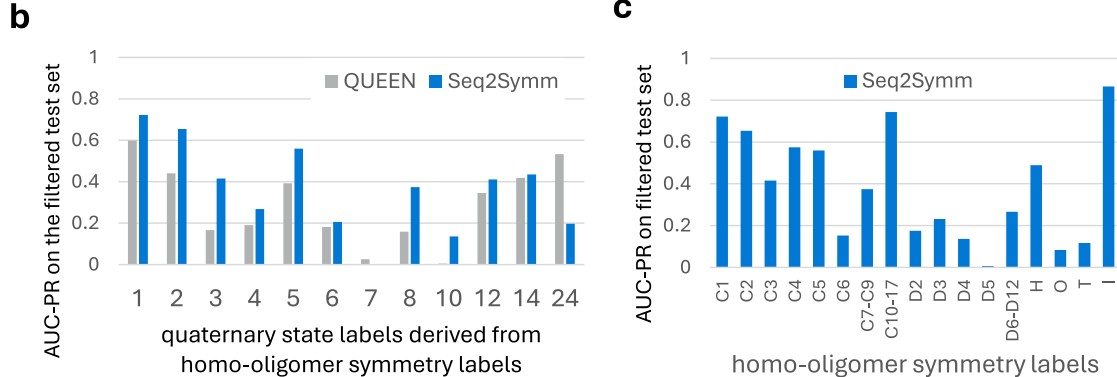

**Fig. 3 | Fine-tuning protein foundation models improves homo-oligomer symmetry prediction and quaternary state prediction. a** Fine-tuning improves model performance across nearly every symmetry group, with the most improvements over pre-trained performance seen in rarer classes. **b** Seq2Symm (ESM2 fine-tuned) outperforms QUEEN, a pre-trained model from prior work on quaternary state prediction **c** The class-wise AUC-PR of Seq2Symm on the filtered test set.

Seq2Symm at 0.643 (detailed results are in Supplementary fig. 12 and the accompanying text). It is worth noting that this setting represents the opposite extreme of the 'no-homology' split.

**Fine-tuning improves performance on minority classes**
Given that fine-tuned models outperform pre-trained models on our test dataset (Fig. 2a), we investigated which homo-oligomer classes explain the difference in performance (Fig. 3a). We find that fine-tuning improves performance on higher-order oligomer symmetries, which are also rarer in the dataset, except in the cases of class C5, where the pre-trained models substantially outperform the fine-tuned model. For the most frequent symmetries in our dataset, such as C1, C2, C3, D2, we

find that pre-trained models are comparable, suggesting that the effects of fine-tuning are most significant for minority data classes. We also note that the benchmarking test split which contains a greater percentage of higher-order oligomer symmetries demonstrates the highest gains from fine-tuning as compared to the Unifold test split, where many of these rarer symmetries are missing (see Supplementary Table 7). These trends are representation-agnostic, as we see a similar behavior with both the MSA-based (ESM-MSA) and the sequence-based (ESM2) models.

To further investigate the benefits of model fine-tuning, we compare the quaternary state predictions of Seq2Symm (our best ESM2 fine-tuned model) with those of a prior approach, QUEEN, which

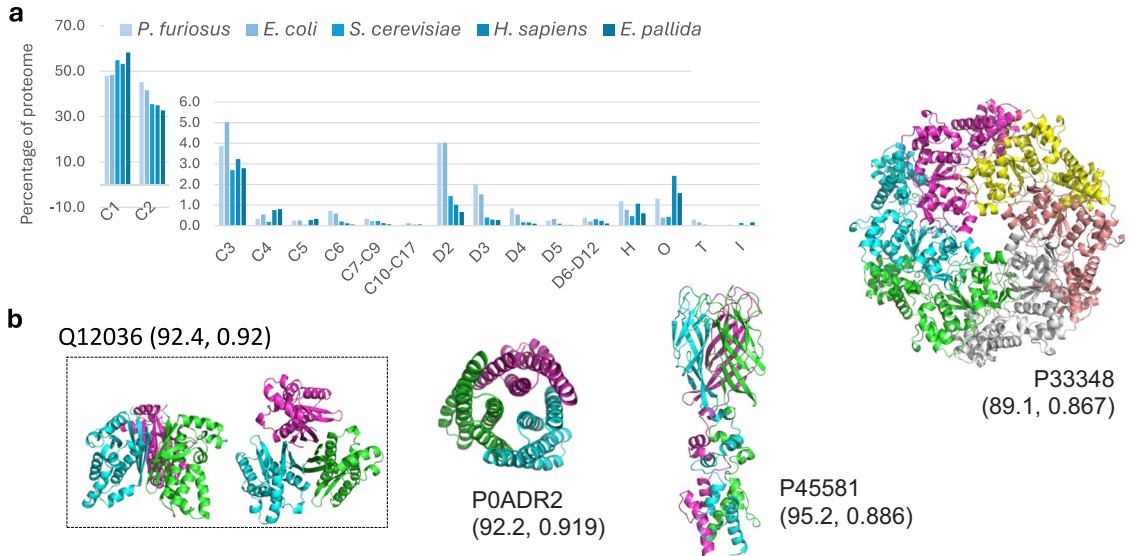

**Fig. 4 | Seq2Symm's rapid predictions enable proteome-wide annotation of homo-oligomer symmetry. a** Proteome-wide distribution of predicted homo-oligomer symmetries in five different organisms depicted as the percentage of all proteins in the proteome reveals a higher percentage of D2 complexes in *P. furiosus* (an archaea) and *E.coli* and a higher percentage of complexes with octahedral symmetry in *H. sapiens* and *E. pallida* (a sea anemone species). **b** Homo-oligomer structures generated using AlphaFold2 based on some of Seq2Symm's homo-oligomer symmetry predictions with (pLDDT, iPTM) scores shown in brackets: Q12036 (*S. cerevisiae*) with 'C3', P33348 (*E. coli*) with 'C6', P0ADR2 (*E. coli*) with 'C3', P45581 (*E. coli*) with 'C3'. High structure quality metrics suggest that Seq2Symm's predictions can aid in generating accurate structures for homo-oligomers.

uses a pre-trained ESM2 model. Specifically, QUEEN uses ESM2 feature embeddings and a supervised head with a single layer (equivalent to logistic regression) to predict one of several quaternary state classes in a multi-class classification setting (see Supplementary Table 8–9, and Supplementary fig. 13 for detailed results). We created a filtered version of our test dataset for comparison purposes by removing proteins that are homologous to any proteins in the QUEEN training set (30% identity, 80% coverage, 1e-3 e-value). This results in a 66% reduction in test structures (from 64,723 to 21,441). Since QUEEN only predicts quaternary state, we convert our test data labels from homo-oligomer symmetries to quaternary states (ex: 'C1' is mapped to 1) producing a many-to-many mapping ('D6' and 'C12' mapped to 12; 'C14' and 'D7' mapped to 14; 'O' and 'I' mapped to 24). This is difficult for some symmetries ('H' can be any of 5,6,7…10,12,13-18,24 and 'T' can be either 12 or 24) and as a result, some structures (848), were discarded due to the lack of a unique label match. Mapping Seq2Symm's output to a single unique quaternary state, for comparison to QUEEN, is complicated by the fact that we coalesce some higher-order symmetries into a single class (Fig. 3c). Nonetheless, Seq2Symm shows superior performance to QUEEN's pre-trained ESM2 model on all quaternary states except 24. This suggests that fine-tuning protein language models for a specific task, rather than simply using their embeddings as inputs to a trainable classifier, can improve performance.

**Rapid predictions of homo-oligomer symmetry across proteomes**

Given the rapid inference time of Seq2Symm, we apply it to five proteomes (*Pyrococcus furiosus, Escherichia coli, Saccharomyces cerevisiae, Homo sapiens*, and *Exaiptasia pallida*) and to a large set of ~3.5 million unlabeled sequences from UniRef50 and metagenomic sources, spanning diverse life forms (see Supplementary Table 10 for details). We show the distribution of various homo-oligomer symmetries, shown as a percentage of the proteome, among the five proteomes in Fig. 4a. We find that the distribution of homo-dimers in our predictions for the four proteomes, 45%, 42%, 35%, 35% in *P. furiosus, E. coli, S. cerevisiae, H. sapiens* respectively, aligns with the findings from[12] (which reported 43%, 44%, 21%, and 21% of the four proteomes

respectively). Across the five proteomes, the prevalence of higher-order symmetries is similar among simpler organisms (*P. furiosus* and *E. coli*) and among the complex organisms (*S. cerevisiae, H. sapiens, E. pallida*), except in the case of proteins with Helical ('H'), Octahedral ('O') and Icosahedral ('I') symmetries. In Supplementary fig. 14, we see the prevalence of multiple homo-oligomeric symmetries per protein in the five proteomes. ~20% of the proteins from *P. furiosus* and *E. coli* have more than one symmetry, while this statistic is ~13% for *S. cerevisiae* and *H. sapiens*.

To analyze Seq2Symm's homo-oligomer symmetry predictions over the ~3.5 million unlabeled proteins, we assign each protein to a superkingdom / kingdom using annotations from UniprotKB and the Taxonomy database. Bacterial proteins constitute 53% of the proteins in this set and the rest come from other organisms (see the 'Overall' bar). In Fig. 5, we show the percentage of proteins in each symmetry class from the various life-forms. We find a significantly higher representation of simpler organisms, mainly bacteria, in the lower-order symmetries C1, C2, C3, D2, D3, with exceptions seen for 'C6', 'H', 'O', 'T'. Reassuringly, viral proteins are overrepresented among 'I' (icosahedral) homo-oligomers. Higher-order C symmetries ('C4','C5', 'C7–C9') see significantly higher representation, and D symmetries ('D4', 'D5', 'D6–D12') are, to some extent more prominent, in higher-order organisms.

To demonstrate the utility of Seq2Symm in generating structures for higher-order oligomeric symmetries, we use Seq2Symm's highest confidence predictions as chain copy number inputs to AlphaFold2 Multimer[22] or ColabFold[23] and generate structures which are depicted in Fig. 4b. By using Seq2Symm, it is possible to bypass an exhaustive search of different homo-oligomer quaternary states as is historically done[12] and instead predict a single homo-oligomer structure based on the output from Seq2Symm (see Supplementary material for details). We compare the computational efficiency of Seq2Symm to a brute-force search using AlphaFold2-multimer for a single protein in Fig. 2f by averaging over ten different C5 homomers ($N = 5$) with an average chain length of 162. The run-time of MoLPC2 will be comparable to that of AF2 multimer $N = 1$ to 7, where several multiplicities are explored before picking the best structure. These are run-times on an Nvidia A100 gpu with 80gb gpu memory. Further, a one-sided paired

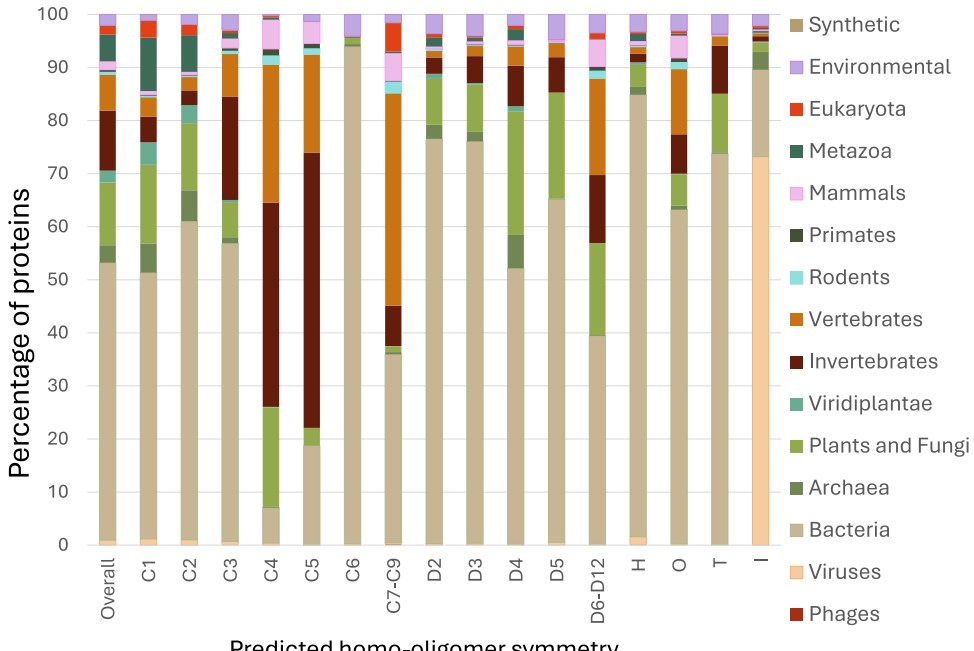

**Fig. 5 | Large-scale predictions reveal patterns across biological kingdoms.** Homo-oligomer symmetry predictions for ~3.5 million unlabeled protein sequences across several biological kingdoms reveal differences in symmetry propensities (e.g., icosahedral symmetry is overrepresented in viruses). For each predicted symmetry, we show the proportion of proteins with that predicted label from each animal kingdom. The leftmost column shows the prevalence of the different kingdoms in the dataset.

t-test that compared predicted confidence scores by AlphaFold-multimer using Seq2Symm's predictions to generate the structure vs. the use of a random prediction ($N = 2$ to 6) to generate the structure found that the structures produced by Seq2Symm's symmetry scored significantly higher in all comparisons (pTM comparison with $p$ value: 6e-4, pLDDT comparison with p-value: 1e-3, piTM comparison with $p$ value of 1e-2).

## Discussion

We describe a rapid deep learning method for accurate prediction of homo-oligomer symmetry. Our approach is computationally efficient and, unlike template-based approaches, does not rely on the availability of symmetry annotations for homo-oligomers on homologous structures. We explore various configurations involving different pFMs and find that the sequence-based ESM2 model upon fine-tuning (Seq2Symm) outperforms template-based homology searches, as well as approaches that finetune or use ESM-MSA or RoseTTAFold2 as pretrained feature extractors.

Seq2Symm outperforms previous methods and pre-trained approaches by an average of approximately 19% on higher-order oligomer symmetries, which are less common in our dataset. Its performance on rare classes, such as C6, C10–C17, and D5, is notable, with AUC-PRs of 0.71, 0.78, and 0.49 on the test dataset, respectively. This suggests that Seq2Symm has likely learned characteristics of these more complex oligomeric symmetry proteins. Our training setup, which involves 1) grouping similar and very rare higher order symmetries into broader classes (e.g., C10–C17 and D6–D12), 2) oversampling the minority classes, and 3) undersampling the majority classes, leads to a reasonable performance on C3, C4, C5, C6, D2, D3, D5, H, and I. We also include a hierarchical loss term for 'coarse classes', grouping all higher-order C and D symmetries into single CX and DX classes, respectively. These strategies have improved the Seq2Symm's performance from 0.50 to 0.52 on the validation set and the ESM-MSA fine-tuned model's performance from 0.27 to 0.45 on the same set. Further improvement is achieved through distillation (see Methods), boosting performance from 0.52 to 0.58 on the validation set.

We note that our model's successful performance is guaranteed in the setting established by our default training regime that defines train/test splits based on a 30% sequence identity cut-off. The performance is expected to be lower in applications involving lesser sequence similarity, as is the case with de novo proteins (as we show in Supplementary fig. 7) or sequences from organisms that might not have related proteins in the PDB (shown by the results in Supplementary figs. 5, 6 on the no-homology data split).

We find that several proteins that have identical sequences and are part of the same MSA, have different labels. For example, the same bovine seminal ribonuclease is assigned different symmetry labels across similar PDB entries (e.g., C2 for `11ba`, 'C2, D2' for `11bg`). This diversity in labels acts as noise for machine learning models and presents a significant challenge. Prior work has suggested that there are discrepancies in oligomer-symmetry annotations, where up to ~10% of biological assembly labels in the PDB are potentially incorrect[24]. Our analysis of the errors made by Seq2Symm reveals that incorrect predictions are often made on proteins that belong to clusters with heterogeneous oligomer symmetries. For instance, in Supplementary fig. 15a which shows the model's predictions for one such heterogeneous cluster, several 'H' symmetry class examples are predicted to be 'C1' or 'CX' (some higher order C symmetry) possibly because the cluster contains many structures with 'C1' and 'C6' labels.

We further report Seq2Symm's performance stratified by main protein structural classes (alpha, beta etc.) and transmembrane proteins in Supplementary Table 11a,b and Supplementary Table 12 respectively. Overall, we find that the models do better more often on structures with mainly-beta folds than on other structural classes.

One key limitation of this work is the high error rate in the confusion region for each class (i.e. in the region with predicted probability of 0.5–0.7) as shown in Supplementary fig. 16. Several avenues exist for improving the performance of models in this study. Currently, all errors are assigned the same penalty during training, but adjusting loss based on class relationships could offer a more nuanced approach, for instance, a misprediction from C3 to C4 being penalized

less than one to C17 or D10. This approach, facilitated by a matrix of size = (# labels) x (# labels), could allow for expressing and optimizing misprediction penalties in a context-aware manner. Our models consider sequence and MSAs as input representations; however, we can also incorporate the structure of the single chain as input (the true structure from PDB where available or the predicted structure otherwise). This is straightforward for models such as RoseTTAFold2, which are designed to input 3D representations. One possibility to extend the applicability of sequence-based models like ESM2, is to use embeddings of 3D structures from structure prediction models, as additional inputs. Another possibility is to fine-tune these models to predict coarse-grained symmetry directly, rather than framing it as a multi-label classification problem.

Lastly, our work predicts homo-oligomer symmetry for a protein, which does not always explicitly encode the quaternary state of the protein (number of subunits), especially in the case of helical ('H') and icosahedral ('I') symmetries. Predicting the symmetry type and quaternary state simultaneously with a single model (e.g., 'H' with 6 chains, 'I' with 180 chains) could improve its utility, possibly without compromising performance.

Nonetheless, Seq2Symm, in its current form, accelerates the modeling of homo-oligomer structural models and the annotation of symmetry groups at the proteome scale. By integrating the output from Seq2Symm with protein structure prediction algorithms, it becomes possible to generate physically realistic 3D structural models of complicated homo-oligomers (Figs. 2c, 4c). Furthermore, Seq2Symm's rapid runtime facilitates the comparison of symmetry group distributions across different species and kingdoms. Thus, Seq2Symm has the potential to become a valuable tool for both proteomic-scale protein structure prediction and comparative analysis.

## Methods

### Datasets

We derive a dataset of 298,771 homo oligomeric labels (that are at the chain-level, such as 11ba_A, 11ba_B) over 129,013 structures from the PDB. For each structure, the global symmetry annotations assigned to all the deposited biological assemblies are considered while defining the homo-oligomer symmetry of the structure. We use all annotations in a multi-label prediction setting and try different approaches to incorporate the multiple labels of a single structure such as using soft labels for all symmetry annotations other than the one from 'biological assembly 1', lower misclassification penalty for annotations from later biological assemblies. We find that treating all labels equally results in a model with the best performance on validation data.

The symmetry annotations in the PDB are assigned by the depositing authors and/or computed by the PISA algorithm[6], which computes various statistics using the deposited atomic structures obtained from X-ray crystallography experiments. PISA uses a scoring function that combines several criteria such as interface contact area, number of interfacial buried residues, salt bridges, disulfide bonds etc[25]. to distinguish the biologically relevant interfaces that define an oligomeric complex from the irrelevant lattice contacts in protein crystals. While there are several other newer tools in the field[26,27], PISA is still considered the gold-standard for estimating the quaternary state[27].

There are 45 different homo-oligomer symmetry labels in our dataset, the most frequent being the 'monomeric' (C1) and the 'cyclic dimeric' (C2) symmetries while higher order symmetry labels are less well represented (Supplementary Table 4). Since certain structures have multiple assemblies, these can have multiple homomer symmetries in our dataset. For example, 6nal[28] has 'C1' and 'C2' labels. There are 17,758 such structures (~6% of the dataset), with some structures having as many as 4 labels and a total of 131 different label combinations (Supplementary Table 4).

### Data splits

To ensure that the test data does not contain homologs of proteins seen during training, we create the train/validation/test splits based on sequence similarity. We use MMSeqs2[29] with a threshold of >30% sequence identity and >80% sequence coverage to cluster the structures, which results in a total of 19,200 clusters. Each cluster is then assigned to one of the train, validation or test splits. This is a more relaxed criterion for clustering proteins as compared to the >80% coverage and >50% sequence identity cut-offs used in the ESM models[15,30], thereby resulting in data splits, where the similarity across splits is much lower. The multi-domain structure of proteins is most likely preserved when using a coverage of >80%.

We select 70% of the clusters to be the training data (13,433 clusters), 10% for the validation split (1860 clusters) and 20% for the test split (3907 clusters). In terms of protein structures, this is equivalent to 205,548 training, 28,509 validation and 64,723 test structures. In MSAs this equates to 49,584 training, 7304 validation and 15,710 test MSAs (one MSA made per unique protein sequence, which results in fewer MSAs due to homologous proteins). All machine learning methods were trained using the same data splits. We discuss the no-homology split in the Supplementary material.

### Evaluation

We use the training split for training all models and the validation split for hyper-parameter and model selection. The test split and other evaluation sets were unseen until the final models were selected and were then used to evaluate final performance.

### UniFold test set

In addition to our curated dataset, we use another completely unseen set of protein structures for the final evaluation of the models, curated from the test set of the UniFold structure prediction model[13], which has 163 structures. After filtering for hetero-oligomer labels we get 96 structures, of which, we were able to construct MSAs for 94 structures using the HHblits algorithm[31] with an e-value cut-off 1e-3 (searching over the following databases: protein sequences from the UniRef30_2023_02[32] version and BFD[33]) and filtered for quality using hhfilter with 90% identity and 75% coverage. We remove proteins that were sequence-similar to our homo-oligomer dataset (30% identity, 80% coverage, 1e-3 e-value); this gives us 83 structures with 85 labels (Supplementary Table 7). All methods are evaluated on these 83 structures as we have both sequence and MSA for these.

### PDB 2024 test set

We curate 152 homo-oligomer structures from the PDB 2024 release (see Supplementary Table 13 for details and Supplementary Table 14 for the dataset).

### De novo test set

To expand the scope of inference and test the transferability of methods to examples, which have highly divergent amino acid sequences from those in the training data, we curated a small test-set of de novo-designed proteins. Additionally, in protein design, an in silico method to screen for oligomeric symmetry prediction would assist in oligomeric design. We collect experimentally resolved symmetric oligomers generated using *hallucination*[34] and *RFdiffusion*[35] and sequences fit with *ProteinMPNN*[36]. Symmetry groups of designs were validated using one or more of the following methods: size-exclusion chromatography (SEC-MALS), negative stain electron microscopy (nsEM), cryo-EM, or X-ray crystallography (See Supplementary Table 15 for the dataset).

### Class imbalance

Our dataset of protein homomer symmetries is heavily class imbalanced due to the high prevalence of certain symmetries such as C1

(monomers) and C2 (cyclic dimers) (Supplementary Table 4). Given the dearth of labels on several higher order symmetry categories, we either prune very rare classes or merge the rarer categories into larger groups. The following classes are merged: C7–C9, C10–C17, and D6–D12.

While the merging of the rarer classes addresses this to an extent, further techniques are needed to adjust for class imbalance during training. We use under-sampling of the majority class, where we under-sample the majority classes C1 and C2 to 70% of their original sizes, followed by oversampling of the minority classes. We over-sample the "extreme minority" classes with fewer than 10,000 protein structures to 5 times their original size (for instance C5 with ~4000 training examples is upsampled to be ~20,000 examples) and the "moderate minority" classes which have more than 10,000 examples to twice their original size. For example, C3 with ~7000 training examples is upsampled to ~14,000 examples. This sampling is done as a pre-processing step prior to training, and is only done on the training split of the data and all models were trained using the same sampled dataset.

### Pre-trained models as feature encoders
The pre-trained models: ESM-MSA, ESM2, and RoseTTAFold2 are used as feature encoders, whereby input proteins are embedded using the hidden layer representations from the neural network models.

The ESM-MSA Transformer uses only the multiple sequence alignment (MSA) of the given protein as input and produces 768-dimensional embeddings for each input residue, which we aggregate by averaging into a single 768-dimensional embedding. To prevent out-of-memory errors during inference, we crop input MSAs by truncating the N-terminal portion of the sequence at 1024 residues (~1% of the structures in our dataset have protein sequences longer than 1024). Further, we select no more than 128 protein sequences per MSA using a greedy selection algorithm based on pairwise Hamming-distances between the protein sequences from the input MSA, as prescribed in the original work.

We obtain 256-dimensional embeddings from RoseTTAFold-2 (RF2), by excluding the 3D track of the model and using as inputs: the MSA of the given protein (1D track), a default structure template (2D track), and averaging the embeddings over the residues of the input protein. The input MSA is cropped to a length of 1024 residues for computational efficiency, by taking a random region of the MSA of length 1024.

The ESM2 model operates on single protein amino acid sequences as input and the embeddings produced by this model have a dimensionality of 1,280. We specifically use the esm2_t33_650 M_UR50D version of the model consisting of 33 layers and 650M parameters that was trained on the UniRef50 database. Analogous to ESM-MSA, we truncate protein sequences longer than 1024 amino acids, by deleting the N-terminal.

Given the embeddings obtained from these pre-trained models as the 'features' for an input protein, we train supervised models for predicting the protein's homomer symmetry using both linear (logistic regression) and non-linear model architectures.

### Fine Tuning protein language models
In addition to using the protein language models as pre-trained feature extractors, we also fine-tune the weights from the original models to adapt to the task of homo-oligomer symmetry prediction. While fine-tuning these models, we do not use the loss functions (such as masked amino-acid prediction or pLDDT, etc.) that were used to train the original models and instead optimize the model for predicting the homomer symmetry. Towards this, we experiment with the following supervised neural network architectures and loss functions.

The number of layers to fine-tune was a hyper-parameter that we picked based on validation set performance. We tried to fine-tune 1, 2, 4, 8 layers from all protein foundation models, and the following

additional options for the number of layers from ESM-MSA and ESM2: 12 and 'all' layers, with gpu memory and compute-time setting the limit on how many layers were possible to fine-tune from each model. We found that there were no gains in performance beyond fine-tuning 2 layers of the model for all three models: ESM2, ESM-MSA and RF2.

### Architecture of the supervised head
**Multilayer perceptron.** This is a simple one- or two-layer feedforward neural network with linear or ReLU activation. We do not incorporate layer normalization or drop-out here as we did not see any changes to the performance on the validation set.

**RobertaLMHead.** The architecture of this module is an extension of the masked language modeling prediction head from ESM. This module begins with a linear transformation, followed by the application of a GELU (Gaussian Error Linear Unit)[37] activation function, introducing non-linearity. A dropout layer, with a configurable rate, is applied post-activation to enhance model robustness. Subsequently, a custom layer normalization is applied (ESM1bLayerNorm) [esm/esm/modules.py at main·facebookresearch/esm·GitHub]. Next, we average over the protein residues, creating a summary representation. Finally, the summary representation is linearly mapped to the output dimension (number of classes) using a dense layer.

**Multitask RobertaLMHead.** We train one supervised head per class, where each head has a RobertaLMHead architecture. There are thus separate parameters for each class, like in a multitask learning setting, with only the protein language model parameters being shared between them.

**Loss functions.** Since our goal is multi-label multi-class classification, we use the binary cross entropy with logits loss function (BCE-WithLogits). BCE with logits treats each class label independently, where for each label, the loss is computed based on the predicted probability and the true label and the total loss is a summation of the independent class-level loss terms, thereby making it possible for an example to have multiple labels.

**Margin loss function.** For each example, we compute a pairwise loss inspired by contrastive learning that constructs pairs of positive and negative labels (all oligomer-symmetries that are not the correct symmetry are considered "negative") and calculate the hinge-loss for each pair as defined below. Given a protein sample $x$ with oligomer symmetry class vector given by $y = [0, 1]^C$ where $C$ is the number of homo-oligomer symmetries, our model $F$ predicts $y' \in \Re^C = F(x)$. We then compute the sample loss as

$$L(y', y) = \sum_{\forall (i, c),\, y'_c \neq 1,\, y'_i \neq 1} \max(0, \, -(y'_c - y'_i) + m) \tag{1}$$

where $m$ is the margin, here 1.0.

The loss in (1) encourages the model to rank the positive labels of an example higher than negative labels by at least the margin. This loss is unlike typical contrastive learning losses, where the positives and negatives are examples, rather than different labels of the same example.

### Template-based prediction of homomer symmetry
We also implement a template matching procedure using HHSearch[7]. The homomer symmetry label for a query structure is assigned based on the homomer labels of the matched template structures, called the "hits". HHSearch sorts the matches by the estimated probability of the matched template to be homologous to the query sequence. Since the estimated probability being greater than 95% indicates that homology is nearly certain, considering hits in this order always starts with the

best matching homologs[7]. In this sorting, we find that top hits are also roughly sorted by the hit-score and the e-value, but not by sequence identity[7]. We assign all labels from the top-*k* matches, which results in a multi-label output. We vary the number of top matches '*k*' that we consider from the returned hits, to get a trade-off between the precision and recall for this approach. To be consistent with the comparison to the machine learning models, we exclude hits to proteins in the test split while evaluating on the test data (we do this analogously while evaluating on the validation data). The other parameters used to run HHSearch are: maximum number of hits of 1000, e-value threshold of 0.001 (see Supplementary material for details). No labels are predicted for proteins where no hits were found. No matches are found for 744 and 1084 proteins from the validation and test sets respectively.

### Distillation

Distillation, also called pseudo-labeling in semi-supervised learning, involves training an initial model on labeled data and using it to assign new "pseudo" labels on a very large unlabeled dataset. These "newly labeled" examples are subsequently used to expand the training dataset, on which a secondary model is trained, which can often generalize better. We use Seq2Symm to generate new labels on our distillation dataset which contains ~7.6 million proteins from UniRef50. We first select proteins satisfying the cut-off of pLDDT > 0.8 and pick one random protein per cluster (where clusters were protein sequence based at 30% identity, 80% coverage, 1e-3 e-value) giving us ~2.8 million input proteins. We run inference on this set to get predicted probabilities per oligomer symmetry class and select all structures that satisfy our class-specific classifier thresholds and exclude C1 or C2 predictions, on account of their over-representation in our gold standard dataset.

### Reporting summary

Further information on research design is available in the Nature Portfolio Reporting Summary linked to this article.

## Data availability

Unless otherwise stated, all data supporting the results of this study can be found in the article, supplementary, and source data files. All data is also available at [https://github.com/microsoft/seq2symm][38] and [https://doi.org/10.5281/zenodo.14681124][39]. Source Data is provided with this paper and available on GitHub[38] and Zenodo[39]. Source data are provided with this paper.

## Code availability

Code to train the model as well as a jupyter-notebook, colab notebook and a python script illustrating how to obtain large-scale predictions are made available in the repository at [https://github.com/microsoft/seq2symm] on GitHub[38] and [https://doi.org/10.5281/zenodo.14681124] on Zenodo[39].

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

## Acknowledgements

We would like to thank Burkhard Rost for his suggestions in improving the analysis in this manuscript. I.R.H. was supported by funding from Bill and Melinda Gates Foundation #OPP1156262. S.S. was supported by the NSF Graduate Research Fellowship under Grant No. 2141064. A.M. was supported by the National Institutes of Health F30 Fellowship (1F30HL162431-01A1). B.B. was supported by the NIH Grant R35GM141861. M.B. was supported by IITP/MSIT (RS-2023-00220628), NRF/MSIT (RS-2023-00210147), and the New Faculty Startup Fund from Seoul National University. D.B. is a Howard Hughes Medical Institute investigator.

## Author contributions

M.B., M.K., I.R.H. contributed to problem formulation and research design. M.K., M.B., I.R.H., A.M., S.S., D.B., B.B. contributed to the methodology, contextualizing the results, planning experiments and analysis. M.B. and I.R.H. collected and processed the datasets. M.K., Y.X., S.S. contributed to the data analysis, M.K. trained many of the models, A.M., S.S. contributed to the baseline training experiments. Y.X. contributed to the multitask training experiments. M.K., A.M., I.R.H., M.B., S.S. drafted most of the manuscript and supplementary material. Y.X. contributed to drafting of the supplementary material, paper review. D.B., J.L.F., R.D. helped with critical paper review and revisions, interpretation of results. D.B., M.B., J.L.F., E.H., R.D. contributed via overall guidance, facilitating collaborations, securing funding and resources, paper review. B.B., G.R.B. contributed via overall guidance, paper review and interpretation of results.

## Competing interests

The authors declare no competing interests.
