## [Transparent Peer Review file · Nature Communications]

Rapid and accurate prediction of protein homo-oligomer symmetry

Corresponding Author: Dr Meghana Kshirsagar

Version 0:

Reviewer comments:

Reviewer #1

(Remarks to the Author)

General comments:

This paper reports a new large protein language model-based method to predict the symmetry of homo-oligomers, which is an import, yet not-well-studied problem. The method not only performs better than existing methods, but also yields some interesting discoveries and applications, including (1) an ESM2 fine-tuned model works better than other language models, (2) using a single-sequence as input works better than using multiple sequence alignments (MSAs) as input, and (3) the predicted symmetries can be used for AlphaFold-multimer to build quaternary structures. The tool runs very fast. It has been applied to predict and analyze the symmetries of millions of proteins in several species / life kingdoms, generating useful hypotheses for the distributions of the symmetries of homo-oligomers in them. Moreover, the strengths and weaknesses of the proposed method were well analyzed. Overall, the method is useful and advances the state of the art. However, there are still some issues below to be addressed to make the method more useful.

Specific comments for revision:

(1) The work tested a very useful application of using the oligomeric symmetries predicted by Seq2Symm as input for AlphaFold-multimer to predict the structures of protein assemblies. This can help users quickly build structures for homo-oligomers without spending a lot of time on exhaustively trying different stoichiometries. To assess how well this approach works, could you compare the AlphaFold-multimer confidence scores of the structures generated for the 296 oligomeric symmetries predicted by Seq2Symm with those generated for randomly selected other symmetries? This comparison can check if the confidence scores of the structures for the symmetries predicted by Seq2Symm are significantly higher than the confidence scores for random symmetries that are most likely incorrect as one would expect. If yes, the comparison can provide further evidence to confirm the good quality of the symmetries predicted by Seq2Symm because the structures for mostly correct symmetries are expected to have higher confidence scores than the incorrect ones.

(2) When I reviewed the manuscript, the GitHub repository of the tool listed in the manuscript was not available. Therefore, I could not check the usability and user-friendliness of the tool.

(Remarks on code availability)

The GitHub repository of the code is not accessible. I reviewed the code and data associated with this manuscript. The code is well written. The datasets are also clearly organized. They are very useful for users to reproduce the experiments. However, this code and data package does not provide documents and examples explaining how to train and test the programs. The authors should provide this information when the code and data are officially released at GitHub.

Reviewer #2

(Remarks to the Author)

Dear Authors,

Thank you for submitting your manuscript for evaluation at the Nature Communications. I have read and evaluated the manuscript and saying it shortly, I find it interesting and that it significantly contributes to the field. While I did not find any

major issues, I would be interested to see your comments on this:

- Did you perhaps evaluate the accuracy of prediction with regard to main protein structural classes (alpha, beta, etc.) and/or other structural features like transmembrane proteins, and perhaps even the extensiveness of subunit–subunit interactions? If yes, it would be interesting to see if there are any differences.
- You define identical units as those with $\geq 95\%$ seq identity over 90% length. Why like this, why not 100% if it is homo-oligomer? If not absolutely identical, then one can only talk about pseudo-symmetry if being very strict about this.

Minor comments:

- Some abbreviations, although common in the field, are not defined in the text, e.g. ESM-MSA, however I believe they should be.
- There are few typing mistakes (e.g. proteins' at line 46, denovo at line 122), and instead of hyphens the en dashes should be used when specifying a range (e.g. C10-C17, line 156).
- Introduction, lines 61–63: the description of cubic symmetry is not well-written since from the current formulation it seems that cubic symmetry includes 3-fold rotational axes.
- The protein name in Fig. 1 is not 11bg, this is the PDB ID of the ribonuclease structure, so please note this clearly.
- Use labels consistently (e.g. PR-AUC vs AUC-PR).

(Remarks on code availability)

Reviewer #3

(Remarks to the Author)

The authors develop Seq2Symm, which takes (fine-tuned) outputs of the ESM2 protein language model and use it as input to a prediction head which outputs the most likely symmetry class of the protein sequence in its quaternary structure. The fine-tuned model is compared against a non-fine-tuned version, as well as using a different model, RoseTTAFold2, to generate the input embeddings, with and without fine-tuning. It is shown that fine-tuning the ESM model with an additional prediction head gives the best results, and the authors claim that it outperforms a more traditional HHsearch-based strategy for determining homomeric state. The method is also claimed to be faster than existing methods for homomeric structure prediction, and this enables the authors to predict the homomeric state of five proteomes and a few million unlabelled sequences from UniRef50.

Comments:

It is concerning that the train-test split is based on a 30% sequence identity threshold, rather than more appropriate assessments of homology (which can extend well below the 30% sequence identity limit, right down to 0%, a fact that has been known since the 80s). Indeed, the authors' own analysis indicates significant overlap in Pfam family content between the training and test sets (Supplementary Figure S10c and d), including in the so-called 'no-homology' split. If I understand correctly, the headline results in the main text are presented using the 'conventional split' wherein 70%(!) of the Pfam families seen in the test set were also seen during training. I appreciate the authors' honesty in clearly including these facts, but I think it should have sent alarm bells ringing that there was this much overlap at the sequence family level, never mind at the structural and stoichiometry level. It means the authors are likely overestimating the model's ability to generalize. The discussion from line 185 onwards in the main text does suggest that data leakage is making the task easier than it would be on truly unseen data. There is also no confidence measure in the predictions that might enable users to know when a prediction is likely to be incorrect.

I would suggest at a minimum that the training and testing is re-done after removing those proteins that overlap. It might even be more appropriate to use structural (rather than sequence family) classification schemes such as ECOD, SCOP or CATH, as distinct Pfams can share the same structure (and possibly the same oligomeric state). If that last part turns out not to be true in the majority of cases of homologous sequences, that should be quantified and reported.

It's unclear why the set of sequences that HHsearch was allowed to call a hit (main text, line 554 onwards) was limited to those less than 30% similar to the query. Surely any practitioner using a homology-based method to predict stoichiometry would consider every hit, especially those most similar to the query? Additionally, ESM2 had access to the entirety of UniRef50 during training (implying a 50% identity cutoff); that information is in the model weights and likely persists after fine-tuning on the conventional-split data set as well.

The authors allude to improved computational efficiency at inference time relative to methods that explicitly predict structure as part of the prediction process such as AlphaFold-Multimer, UniFold-Symmetry, etc., a property which enables proteome-scale prediction of symmetry classes in this paper. However, no comparison of computational cost (either training or inference) relative to these other methods is offered. The only comparisons given are to ESM-MSA and RF2 as evaluated in the study. Indeed, if a user wanted a structure of a homomer, it would be impactful to know how much time would be saved by running Seq2Symm followed by a structure prediction tool with pre-specified stoichiometry, rather than existing, inefficient methods.

In the class of 'inefficient' methods I would probably include the recent MOLPC2 (<https://doi.org/10.1093/bioinformatics/btae329>) which aims to algorithmically determine stoichiometry during the process of building multi-chain structures.

The embedding methods are all limited to a context window of 1024 residues to limit memory usage. Sequences/MSAs are

truncated from the N-terminus in the case of ESM, and in the case of RoseTTAFold2, random crops are used. Why this inconsistency, and why not truncate from the C-terminus? Were other approaches compared? It could be useful to know how many examples in the data set were affected by this truncation.

The GitHub link provided is either non-existent or not public, so I have not been able to review its contents. Code has been provided with the materials for review, but it lacks Readme files, installation instructions, and (perhaps most importantly) neural network weights/checkpoint files needed to run it (or instructions for how/where to get them). As such, I cannot verify the authors' claims/predictions as I cannot run the code, even if I could guess the correct versions of the dependencies needed. Reproducing the results of the paper is therefore impossible unless I re-train the method myself, but as indicated below, there is insufficient detail to do that anyway.

The paragraph describing how many layers from the pLMs were fine-tuned (line 504) is vague; the exact number of layers fine-tuned in each model should be reported.

More details of the training procedure are needed (dropout rates, learning rates, weighting of loss terms, and the like). A description of the compute resources and time needed for training the various models would also be informative.

There are 129013 PDB entries (not 129014 as stated) in the data set (`tail -n +2 homomer_pdbids_hash_clusterid_labels.txt | cut -c 1-4 | sort | uniq | wc -l`); possibly the authors counted the header of this file as well.

(Remarks on code availability)

The GitHub link provided is either non-existent or not public, so I have not been able to review its contents. Code has been provided with the materials for review, but it lacks Readme files, installation instructions, and (perhaps most importantly) neural network weights/checkpoint files needed to run it (or instructions for how/where to get them). As such, I cannot verify the authors' claims/predictions as I cannot run the code, even if I could guess the correct versions of the dependencies needed. Reproducing the results of the paper is therefore impossible unless I re-train the method myself, but as indicated elsewhere, there is insufficient detail to do that anyway.

Version 1:

Reviewer comments:

Reviewer #1

(Remarks to the Author)

The authors did a great job in addressing all my previous comments. The quality of the manuscript has been improved.

(Remarks on code availability)

The code has been released at GitHub under the MIT license. The README file is well written and clearly describes how to train and use the method.

Reviewer #2

(Remarks to the Author)

Dear Authors,

Thank you for submitting the revised version of your manuscript. I see that you have appropriately addressed all my remarks, and I have no objection in publication of the revised version of the manuscript.

Kind regards

(Remarks on code availability)

Reviewer #3

(Remarks to the Author)

I thank the authors for their efforts in addressing my concerns. I particularly appreciated the detailed responses including additional data. A couple of issues remain, however.

The information in Fig S13(a) and (b) shows that 50% of the Pfams in this study have exactly one symmetry class, meaning that for proteins in these families, it is sufficient to know the Pfam to get the symmetry class right. The impact on accuracy across the data considered would depend on the distribution of protein sequences among the various Pfams in question, but maybe that could be something to mention.

I'm confused by the response to Q6. My understanding of the use of HHsearch as a baseline would be to ascertain what predictive performance one could get by using HHsearch to identify highly similar proteins in a sequence database and

using the symmetry classes for those proteins to define that of the target protein (a very sensible thing to do). Given this, I didn't understand why the sequence identity cutoff of 30% was originally used. The new description of the HHsearch procedure which uses HHsearch probabilities to define hits is clear, but this description leaves out whether the 30% sequence identity cutoff is still used.

The authors also show that allowing highly sequence similar hits to be included leads to a higher AUC-PR for HHsearch. That's exactly my point - the notion of information leakage doesn't make much sense for a method like HHsearch which isn't based on an overparameterized machine learning model. Unless I'm mistaken, if I wanted to use HHsearch to predict symmetry class for this task, I would look at the most similar (or highest probability) sequence reported by HHsearch and then simply transfer the symmetry class label if available - that should be the baseline over which we hope to see an improvement when using Seq2Symm.

It's also not clear which sequence database is being searched with HHsearch.

One last nitpick in the acknowledgements: Prof Rost's first name is spelled "Burkhard".

(Remarks on code availability)

The GitHub repository is now available, with some instructions and an IPython notebook that shows how to make predictions. I am able to run the model now, but there are still some issues:

1. The .ipynb file doesn't cover the final part of turning a prediction into symmetry labels. I assume that an `np.argmax` on the 'y_pred' saved in the outputs will do, but it'd be nice to not have to guess.
2. The .ipynb file is nice, but I think users would appreciate a command line-driven .py file that takes arguments in a suitable format.
3. As also noted in a GitHub issue that is open at time of writing, all sequences produce (assuming the approach point 1 above is correct) a C3 class prediction when used with the FASTA loader.
4. The SUPPORT.md file has not been populated yet. It'd be nice to know what level of support users can expect.

Version 2:

Reviewer comments:

Reviewer #3

(Remarks to the Author)

Following the authors' responses and edits to the paper I think I now understand where my confusion with the HHsearch baseline came from:

The queries used for HHsearch are from the same test set used to evaluate Seq2Symm, and so they should be searched against the training set proteins only for a fair comparison. The authors' choice of excluding proteins with $\geq 30\%$ sequence identity to the query (I'm still not completely convinced that this is the right way to define the train-test split, but let's ignore that) was to replicate a situation where all proteins similar to the query from the test set are excluded. This is not quite the same as limiting the database to training proteins only (and as a result the E-values will be affected), but I think it's close enough. I'm only sorry that I didn't connect these dots in the last round of review. The revised text makes this point clear.

(Remarks on code availability)

All requested changes/enhancements to the code and repository have been made.

We would like to thank all reviewers for their insightful comments and constructive suggestions that have resulted in improving this work greatly. We further highlight the disadvantages of template-based methods in performance and structure generation methods like AlphaFold2-multimer in efficiency. We provide point-wise responses and a description of the edits made to the paper, below.

REVIEWER COMMENTS

Reviewer #1 (Remarks to the Author):

General comments:

Q1: This paper reports a new large protein language model-based method to predict the symmetry of homo-oligomers, which is an important, yet not-well-studied problem. The method not only performs better than existing methods, but also yields some interesting discoveries and applications, including (1) an ESM2 fine-tuned model works better than other language models, (2) using a single-sequence as input works better than using multiple sequence alignments (MSAs) as input, and (3) the predicted symmetries can be used for AlphaFold-multimer to build quaternary structures. The tool runs very fast. It has been applied to predict and analyze the symmetries of millions of proteins in several species / life kingdoms, generating useful hypotheses for the distributions of the symmetries of homo-oligomers in them. Moreover, the strengths and weaknesses of the proposed method were well analyzed. Overall, the method is useful and advances the state of the art. However, there are still some issues below to be addressed to make the method more useful.

A: We thank the reviewer for their kind words and hope others find our tools useful.

Q2: The work tested a very useful application of using the oligomeric symmetries predicted by Seq2Symm as input for AlphaFold-multimer to predict the structures of protein assemblies. This can help users quickly build structures for homo-oligomers without spending a lot of time on exhaustively trying different stoichiometries. To assess how well this approach works, could you compare the AlphaFold-multimer confidence scores of the structures generated for the 296 oligomeric symmetries predicted by Seq2Symm with those generated for randomly selected other symmetries? This comparison can check if the confidence scores of the structures for the symmetries predicted by Seq2Symm are significantly higher than the confidence scores for random symmetries that are most likely incorrect as one would expect. If yes, the comparison can provide further evidence to confirm the good quality of the symmetries predicted by

Seq2Symm because the structures for mostly correct symmetries are expected to have higher confidence scores than the incorrect ones.

A: We thank the reviewer for this suggestion. We want to note that confidence scores of structures that are very close in multiplicity are not substantially different (for example, the pTM score value for AlphaFold2 predicted structures for the protein O13511 with N=4 chains is 0.226 while that with N=5 chains is 0.216).

Following the reviewer's idea, we did the following hypothesis test. There were 128 structures that we had generated in our experiments (not 296), to illustrate the application of our approach in conjunction with AlphaFold-multimer. For these, we compare the pLDDT, pTM, piTM values obtained using the symmetry predicted by Seq2Symm to those obtained using a random symmetry, only allowing random symmetries with a multiplicity between 2 to 6, to make the comparison more practical and for computational reasons (as against allowing a multiplicity of 20, which is very likely to have significantly different pLDDT values). We did a one-sided paired t-test to compare the two sets of scores and found those produced by Seq2Symm's symmetry to be significantly higher in all comparisons (pTM comparison with p-value: $6e-4$, pLDDT comparison with p-value: $1e-3$, piTM comparison with p-value of $1e-2$).

Q3: When I reviewed the manuscript, the GitHub repository of the tool listed in the manuscript was not available. Therefore, I could not check the usability and user-friendliness of the tool.

A: We apologize for the delay in the github link being approved for release. It is now public and we have updated the repository to include links to the weights of the models. We have also made available a jupyter notebook to show how the model can be used to make predictions and further details about training a model.

(Remarks on code availability):

Q4: The GitHub repository of the code is not accessible. I reviewed the code and data associated with this manuscript. The code is well written. The datasets are also clearly organized. They are very useful for users to reproduce the experiments. However, this code and data package does not provide documents and examples explaining how to train and test the programs. The authors should provide this information when the code and data are officially released at GitHub.

A: We thank the reviewer for checking the code in detail and apologize for the delay in these updates. We want to inform the reviewers that the availability and usability concerns have been addressed as per their suggestions.

Reviewer #2 (Remarks to the Author):

Dear Authors,

Thank you for submitting your manuscript for evaluation at the Nature Communications. I have read and evaluated the manuscript and saying it shortly, I find it interesting and that it significantly contributes to the field. While I did not find any major issues, I would be interested to see your comments on this.

A. We want to thank the reviewer for their interest and constructive suggestions.

Q1: Did you perhaps evaluate the accuracy of prediction with regard to main protein structural classes (alpha, beta, etc.) and/or other structural features like transmembrane proteins, and perhaps even the extensiveness of subunit–subunit interactions? If yes, it would be interesting to see if there are any differences.

A: We thank the reviewer for this suggestion and now provide AUC-PR performance on the test-set stratified by the following structural classes: Alpha, Beta, Alpha-Beta, Other, Transmembrane. We use CATH superfamily annotations (CATH annotations were available for 75% of the test structures) to determine structural classes such as alpha, beta etc. and UniprotKB annotations to determine whether a protein is a transmembrane protein. These results are tabulated in Supplementary Table S12 and Table S13a, Table S13b. Overall, we find that Seq2Symm does better more often on structures with “mainly beta” folds than on other structural classes.

Q2: You define identical units as those with $\geq 95\%$ seq identity over 90% length. Why like this, why not 100% if it is homo-oligomer? If not absolutely identical, then one can only talk about pseudo-symmetry if being very strict about this.

A: Thank you for your comment regarding our definition of identical units. We chose a threshold of $\geq 95\%$ sequence identity over 90% of the length to balance the need for a large, diverse training dataset with the practical considerations of biological relevance.

While 100% identity might seem more appropriate for homo-oligomers, the slight relaxation in the threshold accounts for the fact that some proteins deposited in the PDB can have different missing regions, even within homo-oligomers. These missing regions do not necessarily impact the functional or structural properties of the oligomer.

Q3. Minor comments:

A. We thank the reviewer for the detailed comments and we have made all the suggested changes.

Reviewer #3 (Remarks to the Author):

Q1: It is concerning that the train-test split is based on a 30% sequence identity threshold, rather than more appropriate assessments of homology (which can extend well below the 30% sequence identity limit, right down to 0%, a fact that has been known since the 80s).

A: We appreciate the reviewer's comment and provide justification for our choice of threshold here and in the main text. We selected a threshold consistent with prior literature, including AlphaFold [Jumper et al. 2021] (Figure 4a in the manuscript), DeepMSA [Zhang et al., 2020], DeepTMHMM [Hallgren et al., 2022]. The threshold of 30% identity at 80% sequence coverage to define “related” proteins leads to stricter training and test splits than some prior works such as ESM1b [Rives et al., 2021], where the authors evaluate the model on UniRef50 clusters. In UniRef50 clusters, each cluster contains sequences that have at least 50% sequence identity and 80% overlap with the longest sequence (seed sequence) in the cluster.

The seminal paper on protein sequence alignments [Rost et al., 1999] found that among all pairs of proteins with sequence identity less than 25%, fewer than 10% of the pairs were homologous and that the 20-30% sequence identity range is considered the “twilight zone” of homology. Further, non-deep learning based methods for structure prediction that rely on protein sequence find that when the evolutionary relationship between the query and template is more distant, typically when the sequence identity is below 30%, the query-template alignment accuracy sharply declines [Kc. D. B. 2017].

We evaluate our models in two settings to gauge its strengths and weaknesses: (a) expanding the biological understanding of homo-oligomers in underexplored proteomes of interest and (b) protein design applications involving *de novo* proteins, which represents an important application that is not the focus of the current work. Based on

the analysis that we present in response to subsequent questions from the reviewer, we can conclude that the 30% sequence identity threshold is reasonable for applications falling in the category (a).

We have added the following line to the main text for clarity:

We call this data split a 'conventional split', since the same criterion has been used in prior work to define data splits in deep learning models such as AlphaFold [11], DeepMSA [37], DeepTMHMM [38].

Q2: Indeed, the authors' own analysis indicates significant overlap in Pfam family content between the training and test sets, including in the so-called 'no-homology' split. If I understand correctly, the headline results in the main text are presented using the 'conventional split' wherein 70%(!) of the Pfam families seen in the test set were also seen during training. I appreciate the authors' honesty in clearly including these facts, but I think it should have sent alarm bells ringing that there was this much overlap at the sequence family level, never mind at the structural and stoichiometry level. It means the authors are likely overestimating the model's ability to generalize. The discussion from line 185 onwards in the main text does suggest that data leakage is making the task easier than it would be on truly unseen data.

A: We thank the reviewer for their comment and apologize for the unclear text and analysis that led to the alarm bells ringing. We understand this concern given recent studies on how protein-ligand interactions or protein-docking models' performance is overestimated due to the data splits. However, this paper focuses on homo-oligomer symmetry, which involves interfaces that are structurally (symmetric and larger) and functionally different from those involved in protein-ligand interactions.

We find that proteins from the same Pfam family can have different homo-oligomer symmetry as protein family annotations summarize structural properties of a protein at a coarse level. For instance, the protein family PF00072 has proteins with the following PDB ids and their corresponding symmetry labels in brackets: 1ojl (C6), 115z (C2, D2), 2fka (T), 1dz3 (C2), 1fc3 (H), 5o8y (D3), 6eo2 (C2, D3). This is true for 3621 PFam protein families, representing 50% of all PFam families seen in our dataset, where two proteins from the same family have different homo-oligomer symmetries as shown in Supplementary Fig S13(a). Fig S13(b) shows that there is uncertainty in the labels in 50% of the protein families.

We can thus conclude that homo-oligomer symmetry is diverse within a protein family and hence, overlap in protein family annotations across the training and test split is not

a meaningful statistic to understand “leakage” from the symmetry prediction perspective.

To further show the relevance of our train/test split for the focus of our work: proteome annotation, we analyze the five proteomes’ (from main text Figure 4) protein family overlap with the two training sets. The average % overlap we see with all the proteomes is 56%, which is much higher than the low 30% overlap seen between the training and test splits of the “no homology” setting. More importantly, note that the overlap of the five proteomes with the “no homology” training split is also similar to that seen with the “conventional” training split. We conclude that the test split that we construct in the “no homology” split is more relevant to applications such as protein design where novel proteins are encountered.

Table Response 1: The number of distinct protein families seen amongst the test proteins of the two data splits and the five proteomes that we consider in this work, using annotations from the PFam database wherever available (66% of the structures in our dataset have PFam annotations).

	Test split (conventional setting)	Test split (non-homology setting)	E.coli.	P.furiosus	H.sapiens	S.cerevisiae	E. pallida
No. of unique protein families	3065	2166	2905	1285	7024	3411	5909

Table Response 2: Overlap between the protein families of the organisms and the training set protein families from the two data splits that we consider is shown below.

	Conventional split (seq-id < 30%, cov 80%)		No-homology split (BLAST e-val 0.1)	
	Overlap with training	% overlap with training	Overlap with training	% overlap with training
Test set (conventional split)	2124	70 %	NA	NA
Test set (no-homology split)	NA	NA	733	30 %
E. coli	1728	59.5 %	1748	60.2 %
P. furiosus	825	64.2 %	838	65.2 %
H. sapiens	3333	47.4 %	3373	48.0 %
S. cerevisiae	1844	54.1 %	1840	53.9 %
E. pallida	3218	54.6 %	3251	55.0 %

Q3: There is also no confidence measure in the predictions that might enable users to know when a prediction is likely to be incorrect.

A: We show the errors made by Seq2Symm across different bins of predicted probability (each bin is of size 0.1) in Supplementary Fig S16. Blue bars show false negative % (for bins 0.0-0.5) and orange bars show the false positive % (for bins 0.5-1.0).

Q4: I would suggest at a minimum that the training and testing is re-done after removing those proteins that overlap.

A: We thank the reviewer for this suggestion and present the following results to address this concern. We analyze Seq2Symm's performance at the level of a protein family, where for each *unseen* protein family, we report the precision and recall. This result is shown in Figure S14c,d (copied in the text below), where we find that out of 589 unseen protein families in our test set, Seq2Symm has 100% recall and 100% precision on 276 and 259 unseen protein families respectively. In Fig S14a,b, we show a similar plot for protein families that were seen during training.

Figure S14. The plots below capture the protein family level performance of Seq2Symm by showing the distribution of the precision and recall averaged over all proteins from a family **(a,b)** performance on test proteins from *seen* protein families **(c,d)** performance on test proteins from *unseen* protein families.

We do an analogous analysis with CATH superfamily overlap (whether a test protein has a CATH annotation seen in the training set or not), shown in Supplementary Table S11b.

In addition to the protein family results, we also show in the table below stratified performance on the test set by splitting test proteins into buckets of BLAST sequence similarity e-value, of a test protein to the most similar training protein. This splits proteins into “easier” and “harder” proteins.

BLAST e-value range of sequence similarity between test protein to the most similar training protein	Number of such test proteins	Seq2Symm's Test AUC-PR
< 1e-5	40983	0.5202945
1e-5 to 1e-3	1788	0.32401357
1e-3 to 0.01	1287	0.26154415
0.01 to 0.1	1475	0.16995251
0.1 to 1.0	2108	0.25974763
> 1.0	4231	0.31555942
no similarity	13151	0.3119903

Q5: It might even be more appropriate to use structural (rather than sequence family) classification schemes such as ECOD, SCOP or CATH, as distinct Pfams can share the same structure (and possibly the same oligomeric state). If that last part turns out not to be true in the majority of cases of homologous sequences, that should be quantified and reported.

A: We opted not to use structural classification for homology because prior work has used sequence-based test/train splits. When considering ECOD, SCOP, or CATH structural annotations, we believe that this would create additional challenges for any case of multi-domain proteins which may have functionally related domains in highly different topologies which would result in more aggressive clustering. Distinct PFams can share the same CATH superfamily annotation, for example: PF00059, PF00084, PF03098 and fifty other protein families are assigned to proteins that belong to the CATH superfamily 2.10.25.10.

Protein family PF00059 (Lectin C-type domain) has several proteins that have very different oligomeric state: 1jzn (C5, D5), 1hup (C3), 1g1t (C1), 1e8i (C2), 3t3a (C2, D2), 3whd (C1, D2) etc. This observation is true for 50% of the protein families in our

dataset, where we find distinct homo-oligomer symmetries amongst proteins with the same PFam family. We want to emphasize that in an analogous manner, proteins from the same CATH superfamily can have different homo-oligomer symmetries.

Q6: It's unclear why the set of sequences that HHSearch was allowed to call a hit (main text, line 554 onwards) was limited to those less than 30% similar to the query. Surely any practitioner using a homology-based method to predict stoichiometry would consider every hit, especially those most similar to the query?

A: We are glad that the reviewer considers this an important aspect of evaluation. Allowing HHSearch to use every hit would be equivalent to creating a randomized split of the dataset where proteins are allowed to have arbitrary sequence similarity across the training and test split. We do not think this setting gives any meaningful understanding of the performance of any method, since a group of structures that have a high sequence identity are often structures of the same protein or at the very least these correspond to very close orthologs or paralogs that have not diverged functionally.

We have clarified our HHSearch evaluation setup better in the write-up now to reflect that we consider hits based on the probability of template match as this is the metric that certainly indicates homology [7]. We do not filter matches based on this probability.

For a more practical comparison as suggested by the reviewer, we now provide results under two different settings:

(1) to provide a “fair” comparison with the ML models, we exclude structures from the test set while evaluating on the test set (and analogously exclude structures from the validation set while evaluating on the validation set). These are now in the updated Figure 1(a,b) of the main text, copied below. The average test-AUCPR of HHSearch is **0.24** if we do not allow access to any test proteins.

(2) For illustration purposes of how easy the task becomes if we do not consider the train/test splits, in Table S14, we show results where, for each test protein, we vary HHSearch’s access to proteins based on the BLAST e-value sequence similarity cut-off of the found matches. As specified in the main text, we get an AUC by varying the number of maximum hits from 1 to 80.

As evident from the “allowed” columns, allowing access to ***only*** very close homologs is sufficient to achieve an AUC-PR of 0.642 due to heavy information leakage. Looking at the “excluded” columns shows the dramatic drop in AUC-PR from disallowing access to very close homologs. We see that the performance does not change much until we reach a lenient BLAST e-value cut-off of 0.01 where it jumps to 0.756 as we allow HHSearch access to all proteins, as suggested by the reviewer. Please contrast these numbers to HHSearch’s low test AUC-PR of **0.24** from (1) above where no test proteins are allowed.

Table S14. Test AUC-PR on the *conventional* train test split for the template-based baseline where HHSearch has **access to all proteins, including all the test proteins, and the only filtering is based on the sequence similarity** of the test protein being evaluated to the proteins found in the “hits”. We can see both the impact of allowing access to test proteins and very close homologs here as we change which proteins are available based on BLAST sequence similarity using the e-value to either include or exclude proteins.

very close homologs allowed (e-val < 1e-5)	very close homologs excluded (e-val > 1e-5)	close homologs are allowed (e-val < 1e-3)	close homologs excluded (e-val > 1e-3)	all homologs are allowed (e-val < 0.01)	all homologs excluded (e-val > 0.01)
0.642	0.402	0.636	0.39	0.756	0.0

Q7: Additionally, ESM2 had access to the entirety of UniRef50 during training (implying a 50% identity cutoff); that information is in the model weights and likely persists after fine-tuning on the conventional-split data set as well.

A: The original ESM2 model is an unsupervised model and was trained to fill in missing amino acids in protein sequences using a masked language modeling based loss function. Predicting one of 20 amino acids to fill in at a given position is a markedly different task than predicting the homo-oligomer symmetry, which represents the quaternary state. While certain sequences in our test set may have been in the ESM2 training set, this is not a concern as our training task was distinctly different from that in ESM2. The unsupervised nature of pLM pre-training allows the model to learn context-based features, akin to what template-based models also match for, which allows these models to generalize and is not a disadvantage or “leakage” concern. Models such as AlphaFold2, RoseTTAFold2 would have some of our test sequences in their training data and also employ a masked amino-acid prediction term in their losses. These models rely on MSAs which use evolutionary information from much larger metagenomic databases, which is letting the models exploit structural similarity across proteins.

Q8: The authors allude to improved computational efficiency at inference time relative to methods that explicitly predict structure as part of the prediction process such as AlphaFold-Multimer, UniFold-Symmetry, etc., a property which enables proteome-scale prediction of symmetry classes in this paper. However, no comparison of computational cost (either training or inference) relative to these other methods is offered. The only comparisons given are to ESM-MSA and RF2 as evaluated in the study. Indeed, if a

user wanted a structure of a homomer, it would be impactful to know how much time would be saved by running Seq2Symm followed by a structure prediction tool with pre-specified stoichiometry, rather than existing, inefficient methods.

In the class of 'inefficient' methods I would probably include the recent MoLPC2 (<https://doi.org/10.1093/bioinformatics/btae329>) which aims to algorithmically determine stoichiometry during the process of building multi-chain structures.

A: We thank the reviewer for this observation and the suggestion for a comparison of computational efficiency. We now discuss MoLPC2 in our introduction. Noting that MoLPC2 uses predictions from the five different AlphaFold-Multimer (AFM) models several times to derive its' final prediction, we can get a lower bound on the computational cost for MoLPC2 using our computational cost statistics for AlphaFold-Multimer. We have hence updated the manuscript as follows to report the numbers for AlphaFold-Multimer in Fig 2f, copied below for easy reference, where we show the total time averaged over 10 proteins for a C5 prediction. The time on the y-axis is in seconds.

Q9: The embedding methods are all limited to a context window of 1024 residues to limit memory usage. Sequences/MSAs are truncated from the N-terminus in the case of ESM, and in the case of RoseTTAFold2, random crops are used. Why this inconsistency, and why not truncate from the C-terminus? Were other approaches compared?

A: We thank the reviewer for noting this. We tried crop sizes smaller than 1024 had a slightly worse performance on the validation set and crop sizes above 1024 lead to out of memory errors in our computational setup. Overall, for both sequence direction as well as crop-size, in order to avoid going against the cropping bias of the original model, we adopted the input processing approaches that were reported to lead to the best results from each model's publications, which happen to be somewhat different in the different models we tried.

Q10: It could be useful to know how many examples in the data set were affected by this truncation.

A: Protein sequences that are longer than 1024 amino acids are affected by this truncation, which constitute around 1% of our dataset (1311 out of a total of 129,014 structures) and remain at 1% in our data splits as well. Since this is a very small proportion of the dataset, the truncation is unlikely to impact training or inference results. We add this information to the manuscript.

Q11: The GitHub link provided is either non-existent or not public, so I have not been able to review its contents.

Code has been provided with the materials for review, but it lacks Readme files, installation instructions, and (perhaps most importantly) neural network weights/checkpoint files needed to run it (or instructions for how/where to get them). As such, I cannot verify the authors' claims/predictions as I cannot run the code, even if I could guess the correct versions of the dependencies needed. Reproducing the results of the paper is therefore impossible unless I re-train the method myself, but as indicated below, there is insufficient detail to do that anyway.

A: We apologize for the delay in the github link being approved for release. It is now public and we have updated the repository to include links to the weights of the models. We have also made available a jupyter notebook to show how the model can be used to make predictions.

Q12: The paragraph describing how many layers from the pLMs were fine-tuned (line 504) is vague; the exact number of layers fine-tuned in each model should be reported.

A. We tried fine-tuning a varying number of layers and found that there were no gains in performance beyond fine-tuning 2 layers of the model. We have updated the text to reflect this observation.

Q13: More details of the training procedure are needed (dropout rates, learning rates, weighting of loss terms, and the like).

A. We add text with this additional information to the Training section of the supplementary material.

Q14: A description of the compute resources and time needed for training the various models would also be informative.

A. We add a description of the compute resources to the Supplementary Material under the Training section. To finetune RoseTTAFold2 we used an Azure VM that had 8 Tesla V100 gpus with 32gb gpu memory. For ESM2 and ESM-MSA we used 4 of these gpus for multi-gpu training using pytorch lightning.

We already provide an estimate of training via the fine-tuning time statistic in Fig 2f for 2 layers. The time taken to fine-tune the full network of ESM2 is 0.23 seconds per protein on one Nvidia A100 gpu with 80gb gpu memory. We could not run our full fine-tuning experiments on the V100 gpus due to gpu memory limits.

Q15: There are 129013 PDB entries (not 129014 as stated) in the data set (`tail -n +2 homomer_pdbids_hash_clusterid_labels.txt | cut -c 1-4 | sort | uniq | wc -l`); possibly the authors counted the header of this file as well.

A. The sharp-eyed reviewer is spot on here. We have corrected this.

Q16: (Remarks on code availability):

The GitHub link provided is either non-existent or not public, so I have not been able to review its contents.

Code has been provided with the materials for review, but it lacks Readme files, installation instructions, and (perhaps most importantly) neural network weights/checkpoint files needed to run it (or instructions for how/where to get them).

As such, I cannot verify the authors' claims/predictions as I cannot run the code, even if I could guess the correct versions of the dependencies needed. Reproducing the results of the paper is therefore impossible unless I re-train the method myself, but as indicated elsewhere, there is insufficient detail to do that anyway.

A. We apologize for the delay in the github link being approved for release. The installation instructions have been made clearer than before. The repo is public and includes links to the weights of the models. We have also made available a jupyter notebook to show how the model can be used to make predictions.

References

Rives, Alexander, et al. "Biological structure and function emerge from scaling unsupervised learning to 250 million protein sequences." *Proceedings of the National Academy of Sciences* 118.15 (2021): e2016239118.

Kc, D. B. (2017). Recent advances in sequence-based protein structure prediction. *Briefings in bioinformatics*, 18(6), 1021-1032.

Chengxin Zhang, Wei Zheng, S M Mortuza, Yang Li, Yang Zhang, DeepMSA: constructing deep multiple sequence alignment to improve contact prediction and fold-recognition for distant-homology proteins, *Bioinformatics*, Volume 36, Issue 7, April 2020, Pages 2105–2112

Jeppe Hallgren, Konstantinos D. Tsirigos, Mads D. Pedersen, José Juan Almagro Armenteros, Paolo Marcatili, Henrik Nielsen, Anders Krogh and Ole Winther (2022). DeepTMHMM predicts alpha and beta transmembrane proteins using deep neural networks. <https://doi.org/10.1101/2022.04.08.487609>

Burkhard Rost, Twilight zone of protein sequence alignments, *Protein Engineering, Design and Selection*, Volume 12, Issue 2, February 1999, Pages 85–94

We want to thank the editor and all the reviewers for their feedback and appreciate their thoroughness in evaluating this work. Below, we address the remaining concerns.

All reviewers:

To give additional evidence of the merits of our approach, we show results on another held-out test set of 151 structures which is the set of new homo-oligomer structures released by PDB in 2024, which we call “PDB 2024”. These structures were selected so that there is no overlap between these and our dataset at 30% sequence identity and 80% sequence coverage. We show the macro averaged AUC-PR (averaged across all symmetries) on this dataset of the four representative approaches in the bar-plot below and include it in Figure 1 in the main paper. Note that the template-based method (HHSearch) has access to our full dataset here (similar to the setting we use for the UniFold-test-set evaluation), while the machine learning models can only access the training split that was used to train them. We find that, in line with previous results, we see a significantly higher AUC-PR with the Seq2Symm models. We see a much lower performance at ~0.25 AUC-PR with both the Template-based method as well as the ESM2 pretrained model (where the output embeddings are used as features on which we train a multilayer perceptron based model).

Reviewer-1:

Thank you for your suggestions and we are glad to have addressed the concerns successfully.

Reviewer-2:

Thank you for your suggestions and we are glad to have addressed the concerns successfully.

Reviewer-3:

1. *The information in Fig S13(a) and (b) shows that 50% of the Pfams in this study have exactly one symmetry class, meaning that for proteins in these families, it is sufficient to know the Pfam to get the symmetry class right. The impact on accuracy across the data considered would depend on the distribution of protein sequences among the various Pfams in question, but maybe that could be something to mention.*

A: Please note that “50% of the protein families” is not equivalent to “50% of the dataset”. We apologize for the text that led to this misunderstanding. Protein families that have a single symmetry class constitute 14.8% of the protein structures from our dataset and 57% of these are C1 proteins i.e. monomers, which are going to be a diverse set of structures. Hence, predicting symmetry based on protein family membership would not be a sound approach.

2. *I'm confused by the response to Q6. My understanding of the use of HHsearch as a baseline would be to ascertain what predictive performance one could get by using HHsearch to identify highly similar proteins in a sequence database and using the symmetry classes for those proteins to define that of the target protein (a very sensible thing to do). Given this, I didn't understand why the sequence identity cutoff of 30% was originally used. The new description of the HHsearch procedure which uses HHsearch probabilities to define hits is clear, but this description leaves out whether the 30% sequence identity cutoff is still used.*

The authors also show that allowing highly sequence similar hits to be included leads to a higher AUC-PR for HHsearch. That's exactly my point - the notion of information leakage doesn't make much sense for a method like HHsearch which isn't based on an overparameterized machine learning model. Unless I'm mistaken, if I wanted to use HHsearch to predict symmetry class for this task, I would look at the most similar (or highest probability) sequence reported by HHsearch and then simply transfer the symmetry class label if available - that should be the baseline over which we hope to see an improvement when using Seq2Symm.

It's also not clear which sequence database is being searched with HHsearch.

A: We apologize for the confusion. In our response to Q6, Table S14 shows an ablation study and these **do not use the 30% sequence identity**. Here, the hits found by HHSearch are either included or excluded based on BLAST sequence similarity to the test protein being evaluated.

The penultimate column of Table S14 shows a setting where HHSearch has access to all proteins (including the validation and test data).

Overall, the results in the table are more of a thought experiment and should not be used as assessment of HHSearch's performance, since the PDB is highly redundant with several structures for the same protein and our dataset also lists each chain from the protein individually (see examples below). Letting HHSearch run without any restriction actually involves substantial information leakage.

Table S14. Test AUC-PR, for the template-based baseline where HHSearch can include or exclude hits based on the ablation criterion listed in each column. We can see the impact of allowing access to very close homologs here. For each test protein, we consider the BLAST sequence similarity of all hits and use the e-value to decide whether to include or exclude hits. We show the e-value cut-off that was used in each column's heading. Note that we don't restrict hits based on data-splits or on any other criteria.

only very close homologs allowed (e-val < 1e-5)	very close homologs excluded (e-val > 1e-5)	only close homologs are allowed (e-val < 1e-3)	close homologs excluded (e-val > 1e-3)	all homologs are allowed (e-val < 0.01)**	all homologs excluded (e-val > 0.01)
0.642	0.402	0.636	0.39	0.756	0.0

** this is equivalent to having access to all proteins in the full dataset that includes the validation and test set, which will have other structures deposited for the same test protein (for example: 102I, 103I, 104I, 107I, 108I etc) and structures from different chains (for example: 104I_A, 104I_B).

Instances of redundant examples (format of the examples is: pbid,sequence,symmetry):

Structures under different control conditions and with slight sequence variations (each one is a separate example in the dataset).

102I_A ,

MNIFEMLRIDEGLRLKIYKDTEGYTIGIGHLLTKSPSLNAAKSELDKAIGRNTNGVITKDEAEKL
FNQDVDAAVRGILRNAKLPVYDSLDAVRRALINMVFQMGETGVAGFTNSLRMLQQKRWDEA
AVNLAKSRWYNQTPNRAKRVITTFRTGTWDAYKNL, C1

103I_A ,

MNIFEMLRIDEGLRLKIYKDTEGYTIGIGHLLTKSPSLNSLDAKSELDKAIGRNTNGVITKDEAE
KLFNQDVDAAVRGILRNAKLPVYDSLDAVRRALINMVFQMGETGVAGFTNSLRMLQQKRWD
EAAVNLAKSRWYNQTPNRAKRVITTFRTGTWDAYKNL, C1

104I_A ,

MNIFEMLRIDEGLRLKIYKDTEGYTIGIGHLLTKSPSLNAAKSAEELDKAIGRNTNGVITKDEAEK

LFNQDVDAAVRGILRNAKLKPVYDSLDAVRRRAALINMVFQMGETGVAGFTNSLRMLQQKRWDE
AAVNLAKSRYWYNQTPNRAKRVITTFRTGTWDAYKNL, C1

104I_B,
MNIFEMLRIDEGLRLKIYKDTEGYTIGIGHLLTKSPSLNAAKSAEELDKAIGRNTNGVITKDEAEK
LFNQDVDAAVRGILRNAKLKPVYDSLDAVRRRAALINMVFQMGETGVAGFTNSLRMLQQKRWDE
AAVNLAKSRYWYNQTPNRAKRVITTFRTGTWDAYKNL, C1

107I_A,
MNIFEMLRIDEGLRLKIYKDTEGYTIGIGHLLTKSPSLNAAKGELDKAIGRNTNGVITKDEAEKLF
NQDVDAAVRGILRNAKLKPVYDSLDAVRRRAALINMVFQMGETGVAGFTNSLRMLQQKRWDEA
AVNLAKSRYWYNQTPNRAKRVITTFRTGTWDAYKNL, C1

108I_A,
MNIFEMLRIDEGLRLKIYKDTEGYTIGIGHLLTKSPSLNAAKIELDKAIGRNTNGVITKDEAEKLF
NQDVDAAVRGILRNAKLKPVYDSLDAVRRRAALINMVFQMGETGVAGFTNSLRMLQQKRWDEA
AVNLAKSRYWYNQTPNRAKRVITTFRTGTWDAYKNL, C1

109I_A,
MNIFEMLRIDEGLRLKIYKDTEGYTIGIGHLLTKSPSLNAAKLELDKAIGRNTNGVITKDEAEKLF
NQDVDAAVRGILRNAKLKPVYDSLDAVRRRAALINMVFQMGETGVAGFTNSLRMLQQKRWDEA
AVNLAKSRYWYNQTPNRAKRVITTFRTGTWDAYKNL, C1

110I_A,
MNIFEMLRIDEGLRLKIYKDTEGYTIGIGHLLTKSPSLNAAKLELDKAIGRNTNGVITKDEAEKLF
NQDVDAAVRGILRNAKLKPVYDSLDAVRRRAALINMVFQMGETGVAGFTNSLRMLQQKRWDEA
AVNLAKSRYWYNQTPNRAKRVITTFRTGTWDAYKNL, C1

164I_A,
MNIFEMLRIDEGLRLKIYKDTEGYTIGIGHLLTKSPSLNAAKSELDKAIGRNTNGVITKDEAEKLF
NQDVDAAVRGILRNAKLKPVYDSLDAVRRRAALINMVFQMGETGVAGFTNSLAMLQQKRWDEAA
VNLAKSRYWYNQTPNRAKRVITTFRTGTWDAYKNL, C1

Structures from different chains:

104I_A,MNIFEMLRIDEGLRLKIYKDTEGYTIGIGHLLTKSPSLNAAKSAEELDKAIGRNTNGVITK
DEAEKLFNQDVDAAVRGILRNAKLKPVYDSLDAVRRRAALINMVFQMGETGVAGFTNSLRMLQQ
KRWDEAAVNLAKSRYWYNQTPNRAKRVITTFRTGTWDAYKNL,C1

104I_B,MNIFEMLRIDEGLRLKIYKDTEGYTIGIGHLLTKSPSLNAAKSAEELDKAIGRNTNGVITK
DEAEKLFNQDVDAAVRGILRNAKLKPVYDSLDAVRRRAALINMVFQMGETGVAGFTNSLRMLQQ
KRWDEAAVNLAKSRYWYNQTPNRAKRVITTFRTGTWDAYKNL,C1

We are not convinced with the reviewer's suggestions of a dramatically different evaluation standard where the baseline can access everything while advocating that a pLM-based approach should be evaluated in a radically stringent regime. Comparing the performance of two approaches in such different settings would not be meaningful and be like comparing apples to unicorns.

We also want to point out that HHSearch is incorporating a different form of supervision through the MSA (constructed using large-scale metagenomic databases), the amino-acid frequencies, gap patterns, PSSMs etc. when it constructs profiles of conserved residues and structural patterns all of which contribute towards learnings that are useful for symmetry prediction.

Given these considerations we now show performance of HHSearch and Seq2Symm on a new train / test split created by MMSeqs clustering using 95% sequence identity at 90% sequence coverage – we call this the “95% setting”. As before, the clusters are split randomly between a training and a test set. In this setup, structures for the same protein are unlikely to be split across the training and test split thereby avoiding the obvious “substantial leakage” concern that we raised above. The Seq2Symm based models are trained on the training split, keeping the same hyper-parameters that were used to train the model on the “conventional split” (i.e. the validation split is not used). Each approach (including HHSearch) only has access to the training split and AUC-PR performance is evaluated on the test split. As before, the template baseline using HHSearch is run by varying the number of hits considered from 1 to 80, to obtain AUC-PR. For HHSearch, no other restrictions are used on sequence identity.

We see the following macro-averaged test AUC-PRs: HHSearch with 0.542, Seq2Symm with 0.643. We show the class-wise AUC-PR on this “95% setting” below. Contrasting the average performance of HHSearch here, which is 0.542, with the numbers in Table S14 (such as 0.756), indicates that those numbers are inflated on account of allowing HHSearch access to duplicates or near-duplicates of the test structure being evaluated.

Please note that it is highly unfair to compare this performance of HHSearch (0.542) where only near duplicates have been removed, to the test AUC-PR of ~0.47 that we see with Seq2Symm on the train/test split where a 30% sequence identity is used to create data splits (shown in Fig 2a). We think that different applications will present a model with inputs that are of varying “difficulty”. Even in an “easy” setting such as the “95% setting” that we show here, a trained deep learning model like Seq2Symm will do better, as we see with a test AUC-PR of 0.643.

Fig R1. Class-wise test AUC-PR

Re: the database, the following text in the supplementary material Page-21 notes the database used: “The database used for the search is PDB version 03 March 2021, which includes all the proteins from our dataset.”

3. Remarks on code availability

The .ipynb file doesn't cover the final part of turning a prediction into symmetry labels. I assume that an `np.argmax` on the 'y_pred' saved in the outputs will do, but it'd be nice to not have to guess.

The .ipynb file is nice, but I think users would appreciate a command line-driven .py file that takes arguments in a suitable format.

As also noted in a GitHub issue that is open at time of writing, all sequences produce (assuming the approach point 1 above is correct) a C3 class prediction when used with the FASTA loader.

The SUPPORT.md file has not been populated yet. It'd be nice to know what level of support users can expect.

A. We are very thankful to the reviewer for these comments. The bug with C3 prediction was due to an indexing issue where, the protein sequence was being passed incorrectly as “U” each time (we use this symbol for unlabeled data). This issue was fixed by the github user who found it. Further, we have made the suggested changes to the code and the SUPPORT.md files.

We are very thankful to all the reviewers and the editor for their helpful comments and suggestions and for their patience with perusing the several revisions. We believe these have greatly improved the manuscript by better positioning our results and by helping present them in a more accurate light.

Reviewer-3:

So far the reviewer has suggested using protein family annotations to construct train/test splits in a previous round of reviews. This is not practical for several reasons, we list two of them here:

(a) a single protein often has several protein family annotations based on the distinct domains it contains, so this is not a reliable way to create train/test splits

(b) protein family membership is a coarser criterion than sequence identity and can result in highly similar proteins getting split across train and test. For example: say protein family PF00552 is in the training split causing `1bai` to be in the training data. Two other structures: `2rsp` and `1mvp` that have up to 50% sequence identity to `1bai` but do not belong to the PF00552 protein family could end up in the test split.

To illustrate generalization across protein families, we already stratify test performance results based on seen and unseen protein families in Supplementary Table S10a.

As cited in our previous response, we use the sequence identity of 30% as it is at the threshold of what is believed to be the twilight zone of homology, since only 10% of protein pairs with sequence identity less than 25% were found to be homologous. We also listed prior works that have used this threshold: AlphaFold [Jumper et al. 2021] (Figure 4a), DeepMSA [Zhang et al., 2020], DeepTMHMM [Hallgren et al., 2022].

The main caveat from using a 30% sequence identity threshold is that our model's performance will be lower in applications involving lesser sequence similarity, as is the case with *de novo* proteins or sequences from organisms that might not have related proteins in PDB. However, in order to illustrate this, we already show results on a *de novo* dataset in Supplementary Figure 7. Further, in Supplementary Figure 6a,b, we show results on another train/test split where we use a BLAST sequence similarity threshold.

We are adding the following new text to the Discussion section of the paper that emphasizes this caveat:

We note that our model's successful performance is guaranteed in the setting established by our default training regime that defines train/test splits based on a 30% sequence identity cut-off. The performance is expected to be lower in applications involving lesser sequence similarity, as is the case with *de novo* proteins (as we show in Supplementary Figure 7) or sequences from organisms that might not have related

proteins in the PDB (shown by the results in Supplementary Figure 6a,b on the no-homology data split).

Further, in order to show that we evaluate HHSearch in a typical use setting, we include the results from the “95% sequence identity setting”, where we compare the AUC-PR with Seq2Symm. The following text was added to the manuscript.

Finally, to get a fair assessment of template-based methods, we design a setting to simulate the “typical” manner of applying HHSearch-like methods that are not “trained”. Using a sequence identity threshold of 95% at a coverage of 90%, we create a data-split where the test set does not have identical or near-identical structures to the training set. We evaluate both HHSearch and Seq2Symm (a model trained on this “95% seq-id” training split) on this test set. We find that HHSearch has a macro-averaged test AUC-PR of 0.542 with Seq2Symm at 0.643 (detailed results are in Supplementary Figure 12 and the accompanying text). It is worth noting that this setting represents the opposite extreme of the ‘no-homology’ split.